# Oriented clonal cell dynamics enables accurate growth and shaping of vertebrate cartilage

Marketa Kaucka[1,2], Tomas Zikmund[3], Marketa Tesarova[3], Daniel Gyllborg[4], Andreas Hellander[5], Josef Jaros[6], Jozef Kaiser[3], Julian Petersen[2], Bara Szarowska[2], Phillip T Newton[1], Vyacheslav Dyachuk[7], Lei Li[1], Hong Qian[8], Anne-Sofie Johansson[8], Yuji Mishina[9], Joshua D Currie[10], Elly M Tanaka[10], Alek Erickson[11], Andrew Dudley[11], Hjalmar Brismar[12], Paul Southam[13], Enrico Coen[13], Min Chen[14], Lee S Weinstein[14], Ales Hampl[6], Ernest Arenas[4], Andrei S Chagin[1,15], Kaj Fried[7]*, Igor Adameyko[1,2]*

[1]Department of Physiology and Pharmacology, Karolinska Institutet, Stockholm, Sweden; [2]Center for Brain Research, Medical University Vienna, Vienna, Austria; [3]Central European Institute of Technology, Brno University of Technology, Brno, Czech Republic; [4]Unit of Molecular Neurobiology, Department of Medical Biochemistry and Biophysics, Karolinska Institutet, Stockholm, Sweden; [5]Department of Information Technology, Uppsala University, Uppsala, Sweden; [6]Department of Histology and Embryology, Medical Faculty, Masaryk University, Brno, Czech Republic; [7]Department of Neuroscience, Karolinska Institutet, Stockholm, Sweden; [8]Department of Medicine, Karolinska Institutet, Stockholm, Sweden; [9]Department of Biologic and Materials Sciences, University of Michigan School of Dentistry, Ann Arbor, United States; [10]Center for Regenerative Therapies, Technische Universität Dresden, Dresden, Germany; [11]Department of Genetics, Cell Biology and Anatomy, University of Nebraska Medical Center, Omaha, United States; [12]Science for Life Laboratory, Royal Institute of Technology, Solna, Sweden; [13]John Innes Centre, Norwich, United Kingdom; [14]National Institute of Diabetes and Digestive and Kidney Diseases, National Institutes of Health, Bethesda, United States; [15]Institute for Regenerative Medicine, Sechenov First Moscow State Medical University, Moscow, Russia

*For correspondence: kaj.fried@ki.se (KF); igor.adameyko@ki.se (IA)

Competing interests: The authors declare that no competing interests exist.

**Abstract** Cartilaginous structures are at the core of embryo growth and shaping before the bone forms. Here we report a novel principle of vertebrate cartilage growth that is based on introducing transversally-oriented clones into pre-existing cartilage. This mechanism of growth uncouples the lateral expansion of curved cartilaginous sheets from the control of cartilage thickness, a process which might be the evolutionary mechanism underlying adaptations of facial shape. In rod-shaped cartilage structures (Meckel, ribs and skeletal elements in developing limbs), the transverse integration of clonal columns determines the well-defined diameter and resulting rod-like morphology. We were able to alter cartilage shape by experimentally manipulating clonal geometries. Using in silico modeling, we discovered that anisotropic proliferation might explain cartilage bending and groove formation at the macro-scale.

## Introduction

Cartilage is an essential skeletal and supportive tissue in our body. The shape and size of each cartilage element results from complex developmental processes; mesenchymal cells initially condensate, differentiate into chondrocytes, and then an orchestrated growth of the entire structure occurs (*Goldring et al., 2006*). Often, cartilage plays an important role as a developmental intermediate, such as during the endochondral growth of the long-bones (*Mackie et al., 2008*). Cartilage elements vary widely in their shapes: they may be simple shapes like rods or bars (Meckel, cartilage templates of the future long bones and ribs) or sheet-like structures (in the head), but can be extremely complicated with a huge number of irregular shapes (for instance, in the inner ear or pelvis). The geometrical properties of cartilage elements must be fine-tuned during the growth because cartilage provides indispensable structural support to the body during development. Yet, how this is achieved despite drastic changes in size is unclear.

After early cartilage forms from mesenchymal condensations, growth typically occurs in all dimensions. However, the diversity of cell dynamics controlling precise early growth and shaping is not well studied. At the same time, the late growth of long rod-shaped cartilage elements in limbs is achieved through a mechanism of endochondral ossification that includes oriented cell dynamics in growth plate-like zones (*Vortkamp et al., 1996*). In the germinal zone of a growth plate, chondrocytes proliferate and produce progenies that form long streams oriented along the main axis of the forming skeletal element. Inside such streams, chondrocytes undergo flattening, oriented cell divisions and hypertrophy before dissipating and giving place to the forming bone (*Nilsson et al., 2005*), a process which is controlled by many signals (*Kronenberg, 2003*). This cell dynamic enables efficient extension of the skeletal element in a specific direction that coincides with the orientation of cell divisions in the proliferative zone (*Abad et al., 2002*). Growth plate disorders may result in dwarfism and other illnesses (*De Luca, 2006*).

Some parts of the cartilaginous skull (*e.g.* the basisphenoid of the chondrocranium) also undergo endochondrial ossification in synchondroses, and significant growth of the cranial base is achieved through a similar mechanism (*Hari et al., 2012*; *Wealthall and Herring, 2006*).

Synchondroses are mirror-image growth plates arising in the cranial base, which primarily facilitate growth in the anterio-posterior direction (*Kettunen et al., 2006*; *Laurita et al., 2011*; *Nagayama et al., 2008*; *Young et al., 2006*). Disorders in the development of synchondroses severely impact the elongation of the cranial base and often result in short-faced mutants and a general decrease of the cranial length (*Ford-Hutchinson et al., 2007*; *Ma and Lozanoff, 1999*). Insufficient or abnormal development of a cartilage element is one of the reasons for human craniofacial pathologies, providing a connection between the chondrocranium and facial bone geometry, size and placement (*Wang et al., 1999*).

The growth mechanism operating in growth plates and synchondroses involves the transformation of the cartilage into the bone. Since growth plates or synchondroses are oriented towards a specific direction, the expansion of a cartilage in other dimensions is not clear from the mechanistic point of view and requires further investigation. For example, although it is well known that the mouse chondrocranium develops as 14 independent pairs of cartilage elements that form one united structure, the logic behind further shaping and scaling remains unclear (*Hari et al., 2012*). How these initially separated large cartilaginous elements form, grow and fine-tune their geometry, thickness and smoothness during development is still not completely understood. We hypothesized that accurate cartilage growth might require alternative cell dynamics that do not involve hypertrophy, ossification or growth plates.

Such alternative cell dynamics may also contribute to the accuracy of scaling during cartilage growth. Scaling is a process of growth that maintains both the shape and the proportions of the overall structure. In nature, scaling often involves sophisticated principles of directional growth and a number of feedback mechanisms (*Green et al., 2010*). For instance, during bird development, the diversity in beak shape is constrained by the dynamics of proliferative zones in the anterior face (*Fritz et al., 2014*). Furthermore, scaling variations of beaks with the same basic shape result from signaling that controls the growth of the pre-nasal cartilage and the pre-maxillary bone (*Mallarino et al., 2012*). Indeed, in order to accurately scale a pre-shaped 3D-cartilaginous template both local isotropic and anisotropic cell dynamics may be required.

To assess changes in the complete 3D anatomy of the face following cellular-level mechanistic studies we used a variety of approaches including micro-computed tomography (μ-CT), genetic tracing with multicolor reporter mouse strains, multiple mutants and mathematical modelling.

Most importantly, we reveal here how oriented clonal behavior in the chondrogenic lineage controls the overall geometry of the cartilage elements, and show that this geometry can be manipulated with molecular tools at various levels.

## Results

Cartilage elements form and grow in all parts of the vertebrate body. The developing face provides a remarkable variety of cartilage geometries and sizes and, therefore, may serve as a sophisticated model system to study the induction of complex cartilaginous structures.

The developing cartilaginous skull, the chondrocranium, displays a very complex geometry of mostly sheet-like cartilages that result from coordinated anisotropic growth in all dimensions. Such expansion of sheet-like cartilaginous tissue during embryonic development involves several mechanisms that were proposed in the past, including the formation and growth of cartilage at synchondroses, as well as at the apical growth zone.

To understand the changes in dimensions of chondrocranium growth at major developmental stages, we took advantage of 3D reconstructions using μ-CT enhanced with soft tissue contrasting (*Figure 1*). This approach enables the identification of various tissues and cell types in the embryo based on differential uptake of tungsten ions. We validated the μ-CT visualization of embryonic cartilage by directly aligning stained histological sections with the 3D models (*Figure 1—figure supplements 1* and *2*).

We analyzed the expansion of the chondrocranium due to synchondroses and found that despite a significant anterio-posterior elongation, synchondroses cannot entirely explain the growth dynamics in all directions: anterio-posterior, latero-medial and dorso-ventral vectors of growth (*Figure 1A–C*). Specifically, we found a complete absence of synchondroses and other endochondrial ossifications in the growing nasal capsule, even at the earliest postnatal stages, while membranous ossifications appeared well developed. The stereotypical clonal cell dynamics found in synchondroses (*Figure 1D–I*) did not appear during the development of the nasal capsule. Therefore, during the entire embryonic development, chondrocranium growth and shaping is largely aided by additional and unknown mechanisms of growth.

To investigate another possible mechanism of growth, we examined the apical growth zone of the nasal capsule. To understand growth dynamics there, we birth-dated different regions of the facial cartilage using genetic tracing in *Col2a1-CreERT2/R26Confetti* and *Sox10-CreERT2/R26Confetti* embryos (*Figure 2—figure supplement 1*). Both *Col2a1-CreERT2* and *Sox10-CreERT2* lines recombine in committed chondrocyte progenitors and in mature chondrocytes. 3D analysis following tamoxifen injections at different developmental stages allowed us to identify the parts of the cartilage that develop from pre-existing chondrocytes and the regions generated from other cellular sources (*Figure 2*, *Figure 2—figure supplement 1*). As an example, after genetic recombination induced at E12.5, locations with high amount of traced cells show structures that come from pre-existing cartilage, whereas areas comprising from non-traced cells present structures originating from de novo mesenchymal condensations. We discovered that important and relatively large geometrical features are produced from waves of fresh mesenchymal condensations induced directly adjacent to larger pre-laid cartilage elements between E13.5 and E17.5: this includes the frontal nasal cartilage, nasal concha, labyrinth of ethmoid and, consistent with previous suggestions, cribriform plate (*Figure 2C* and *Figure 2—figure supplement 1A–K*). These results cannot be safely inferred from 2D traditional histological atlases because of the complex geometry. Our results are complementary to the findings of McBratney-Owen and Morris-Key with coworkers, who demonstrated that the complete chondrocranium (including skull base) develops from 14 pairs of early independently induced large cartilaginous elements that fuse together during later development (*McBratney-Owen et al., 2008*). Here, we demonstrated how new adjacent mesenchymal condensation can increase the geometrical complexity of a single solid cartilaginous element.

To substantiate our results, we took advantage of *Ebf2-CreERT2/R26Tomato* transgenic mouse line that can genetically label only a few selected patches of early mesenchyme in the cranial region. We wanted to test if some of these labelled mesenchymal patches can undergo chondrogenesis

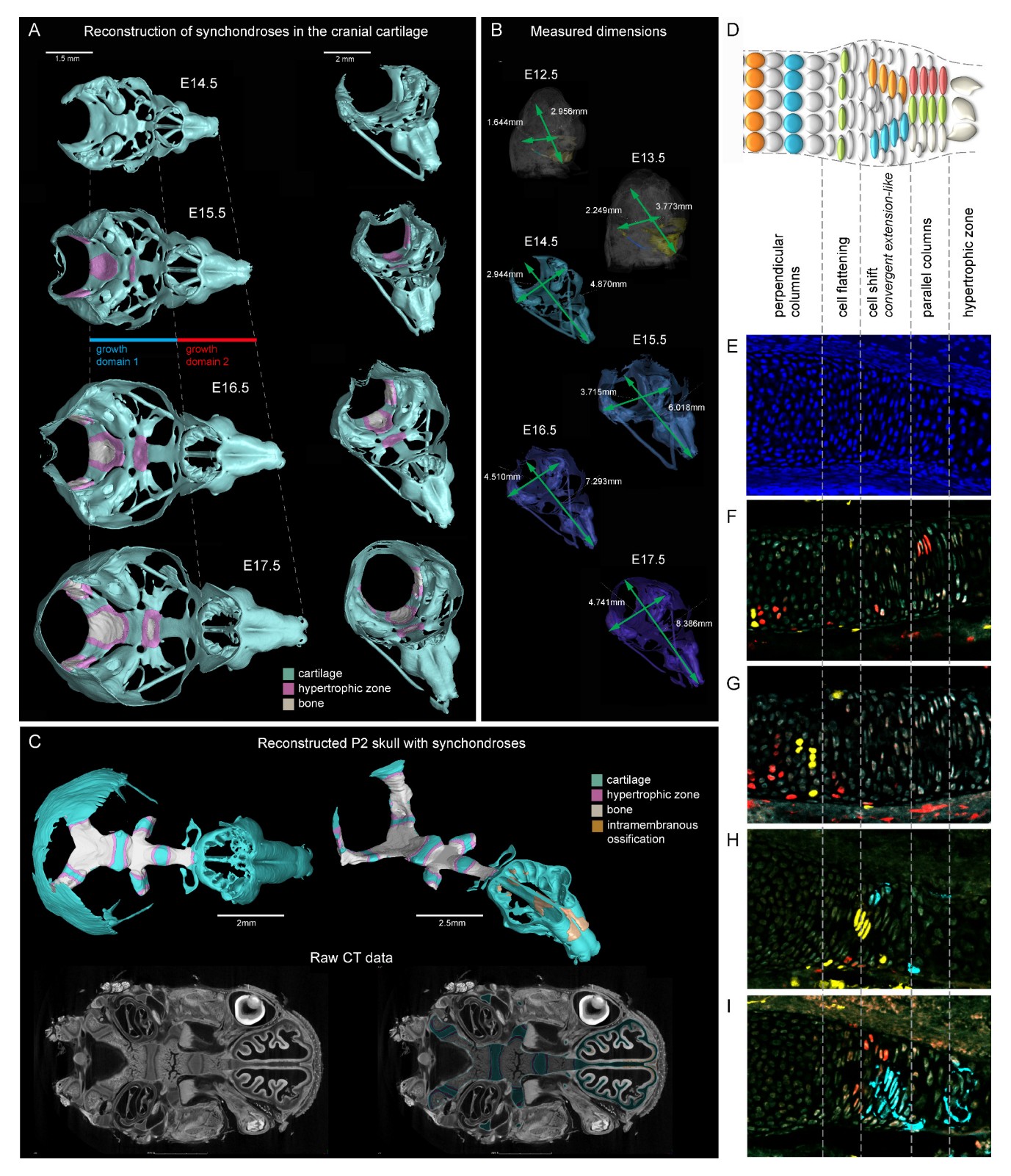

**Figure 1.** Visualizations of endochondrial ossifications in the chondrocranium during development. (**A**) 3D models of chondrocrania with visualized bone and hypertrophic cartilage. Note the absence of endochondrial ossifications in the nasal capsule between E14.5–17.5. Intramembraneous ossifications are not shown. (**B**) Width and length of the chondrocranium in E12.5–17.5 stages. (**C**) P2 stage model with visualized bone formation, hypertrophic zones and intramembraneous ossification in the nasal capsule. Clipping planes are applied for better visualizations of synchondroses.
*Figure 1 continued on next page*

Figure 1 continued

Corresponding raw CT data are presented in the lower part. (D–I) Clonal genetic tracing in synchondroses with *Sox10CreERT2/R26Confetti*; injected at E12.5 and collected at E17.5. (D) Schematic of synchondroses, (E) DAPI stained nuclei, (F–I) different clonal arrangements in various zones of progressing synchondroses.

The following figure supplements are available for figure 1:

**Figure supplement 1.** Histological confirmation of µ-CT results.

**Figure supplement 2.** Immuno- and histological validation of cartilage contrasting obtained from µ-CT analysis and subsequent 3D modelling.

independently and much later than most of the chondrocranium structure. If that would be the case, we could expect the formation of very sharp borders between the labelled and non-labelled cartilage due to the fusion of newly produced labelled cartilage with the old unlabeled one. If the local cartilage would form from labelled and unlabeled mesenchyme at the same time, the border would not form due to mesenchymal clone mixing that we observe when we label early neural crest. We injected *Ebf2-CreERT2/R26Tomato* animals with tamoxifen at E12.5 and analyzed the embryos at E17.5 (*Figure 2—figure supplement 2*). As a result, we discovered that the cartilage element connecting the inner ear with the basisphenoid was genetically traced, and demonstrated a very sharp border with non-traced cartilage (*Figure 2—figure supplement 2C–D*). µ-CT data confirmed that this element develops entirely after E14.5 from newly formed mesenchymal condensations adjoining the chondrocranium (*Figure 2—figure supplement 2A–B*), and this might be related to differential regulation at the neural crest-mesodermal border (*McBratney-Owen et al., 2008*; *Thompson et al., 2012*). At the same time, the main structure of the chondrocranium is expanded in a very precise and symmetrical way due to unknown cellular and molecular mechanisms that cannot be explained by the freshly induced condensations, the apical growth zone, or even cell dynamics in synchondroses. Our µ-CT results (*Figure 2*) show that various parts of the chondrocranium develop due to the growth of pre-existing cartilage not involving ossifications, while only additional features are induced in waves as de novo mesenchymal condensations that fuse with the main element during their maturation or expand in the process of ossification.

We further focused on the developing nasal capsule because its growth does not involve synchondroses while the apical growth zone and adjoining mesenchymal condensation only partly provide for the growth and shaping modifications.

The results obtained from comparisons of cartilaginous nasal capsules from different developmental stages showed that the shape of the structure is generally established by E14.5 (*Figure 3*, *Figure 3—figure supplement 1*, *Figure 3—figure supplement 2*, *Video 1*). Nevertheless, from E14.5 until E17.5 the cartilaginous nasal capsule is accurately scaled up with significant geometrical tuning (*Figure 3A–B*). Previous knowledge suggests that the underlying growth mechanism should be based on appositional growth of the cartilage during its transition to bone (*Hayes et al., 2001*; *Li et al., 2017*), however, numerous facial cartilages never ossify, but continue to grow.

Tomographic reconstructions of sheet-shaped cartilage elements in the nasal capsule revealed extensive expansion of the cartilage surface area and overall volume (*Figure 3E–F*). Surprisingly, the thickness of the cartilaginous sheets did not change as much as the other dimensions during nasal capsule growth (*Figure 3C–F*, *Figure 3—figure supplement 1*, *Video 1*). Thus, the sheet-shaped cartilage expands mostly laterally (within the plane) during directional growth. Therefore, we expected that clonal analysis of the neural crest progeny (with *Plp1-CreERT2/R26Confetti*) and of early chondrocytes (with *Col2a1-CreERT2/R26Confetti* or *Sox10-CreERT2/R26Confetti*) would reveal clonal units (so called clonal envelopes) oriented longitudinally along the axis of the lateral expansion of the cartilage. Surprisingly, and contrary to this, clonal color-coding and genetic tracing demonstrated transversely oriented clones represented by mostly perpendicular cell columns or clusters formed by traced chondrocytes (*Figure 4*, especially A-C, *Figure 4—figure supplement 1*).

To understand this process more in depth, we started with genetic tracing of the neural crest cells and their progeny in the facial cartilage with *Plp1-CreERT2/R26Confetti* (tamoxifen injected at E8.5). Clonal analysis and color-coding of neural crest-derived chondrogenic and non-chondrogenic ectomesenchyme showed intense mixing of neural crest-derived clones in any given location (*Figure 4A–*

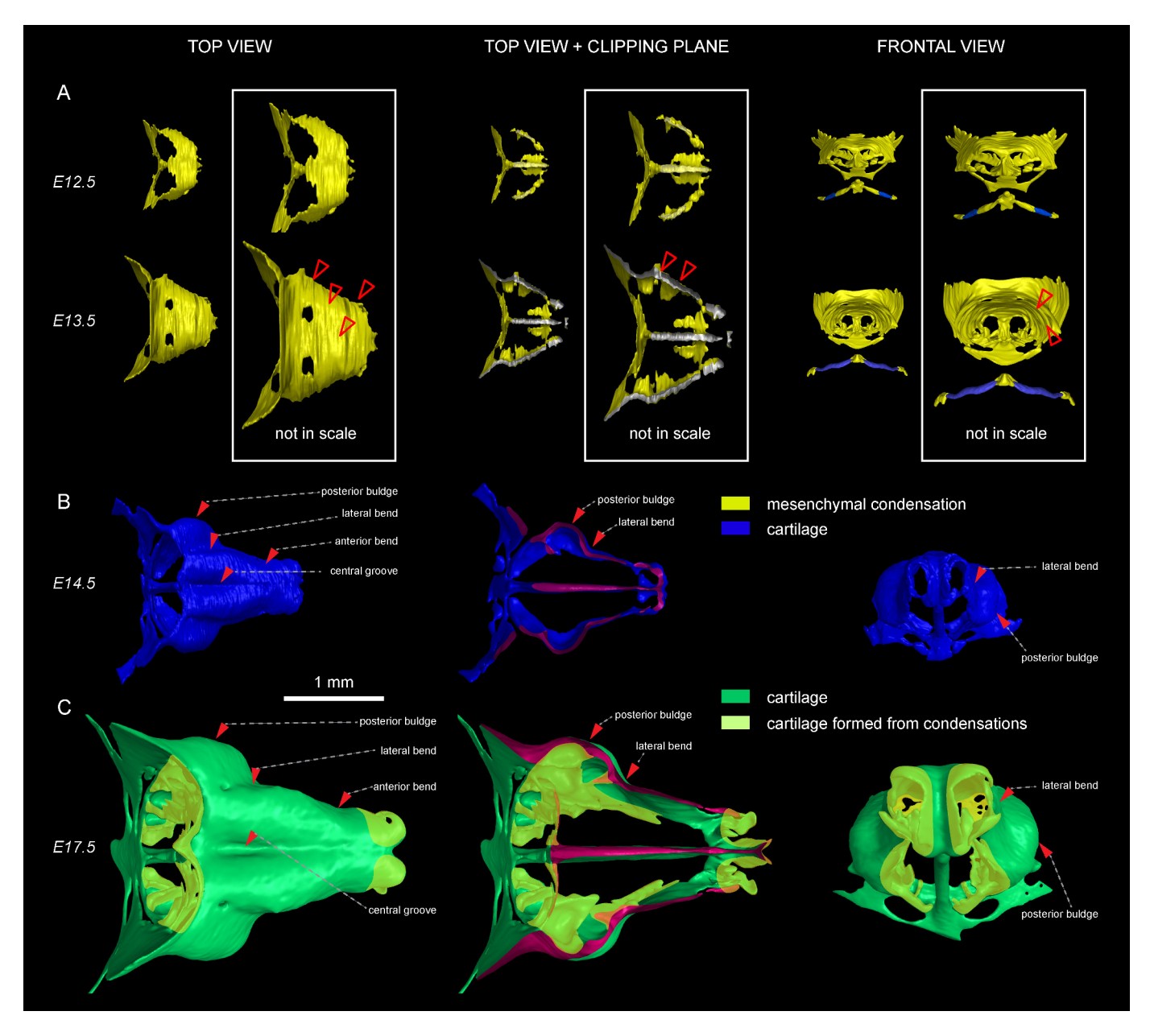

**Figure 2.** Induction of initial shape and geometrical transformations of the facial chondrocranium. (A–C) μ-CT-based 3D reconstruction of chondrogenic mesenchymal condensations and cartilage in the face of E12.5, E13.5, E14.5 and E17.5 embryos. (A) Mesenchymal condensations (yellow) segmented from E12.5 and E13.5 embryos and presented in frontal and top projections. Note that the basics of the facial chondrocranium are already established at the stage of mesenchymal condensations during the early development, while general geometry and fine details are tuned during further transformations. (B–C) Top, clipping plane + top and frontal projections of E14.5 (B) and E17.5 (C) developing facial chondrocranium. (C) Yellow color highlights the results of cartilage birth-dating experiments and shows the areas produced from de novo mesenchymal condensations that appear in successional waves after the primary cartilage (shown in green) is produced at previous stages (E14.5). Note that the shape of the facial chondrocranium develops as a result of incremental formation and additive fusion of new mesenchymal condensation with pre-existing cartilage. Red arrows indicate areas of cartilage which bend at later developmental stages (B,C) and red-outlined arrows indicate the same areas within the mesenchymal condensations at E13.5, prior to bending (A, bottom).

The following figure supplements are available for figure 2:

**Figure supplement 1.** Genetic tracing serves as a tool for birth-dating of the cartilage during the embryonic development.

*Figure 2 continued on next page*

*Figure 2 continued*

**Figure supplement 2.** Genetic tracing based on Ebf2-CreERT2/R26Tomato serves as an indicator for structures developed from late mesenchymal condensations.

*C*, *Figure 4—figure supplement 1D–G*) (*Kaucka et al., 2016*). At the same time, chondrogenic ectomesenchyme demonstrated the presence of transversely oriented doublets of genetically traced and also EdU-labeled cells already at E13.5 (*Figure 4—figure supplement 1G* (inserts) and H). Next, analysis of neural crest progeny in established cartilage highlighted the presence of perpendicularly oriented clonal doublets and columns (*Figure 4A–C*). Further analysis of EdU incorporation and genetic tracing with chondrocyte-specific *Col2a1-CreERT2/R26Confetti* and *Sox10-CreERT2/R26Confetti* lines confirmed the existence of transversely oriented products of cell proliferation in the mature (E14.5-E17.5) cartilage (*Figure 4D–F* for EdU, *Figure 4G,H–L* and *Figure 4—figure supplement 2* for lineage tracing). These results imply that cells in the sheet-shaped cartilage do not allocate daughter cells in lateral (longitudinal) dimensions as would be intuitively expected.

Thus, simple lateral or unidirectional proliferation cannot account for the accurate scaling of the sheet-shaped cartilage in the face. Instead, the cartilage development from chondrogenic condensations is achieved by a cellular mechanism that involves intercalation of columnar clonal units.

It was unclear to us why column-like structures, and no other shapes, are integrated into the sheet-shaped cartilage and how the fine surface is maintained during this mechanism of growth. To better understand possible mechanisms of accurate sheet-shaped cartilage surface development we modelled individual cell dynamics, in silico in 4D (3D + time) (*Figure 5*) (*Hellander, 2015*). We used this modelling to address two questions: firstly, under what conditions are clonal columns observed? Secondly, how is the sheet-like shape achieved by polarized or non-polarized cell divisions of single-cell thick layers and what are the controlling mechanisms? We tested a group of variables including: cell division speed, allocation of daughter cells in random- or defined directions, orientation cues in the tissue (equivalent to molecule gradients), as well as pushing/intercalating of the daughter cells during proliferation. We qualitatively compared the results from in silico simulations to our experimental clonal analysis from various genetic tracing experiments, in order to identify conditions in the model that were compatible with patterns observed in vivo.

The results of the mathematical modelling suggested that the clonal dynamics observed in natural conditions requires polarity cues in the system, specifically, a two-sided gradient of signals would be required to precisely fine- tune cartilage thickness (*Figure 5A–J*). At the same time, some yet to be identified mechanism controls the average number of cell divisions in a column, further controlling columnar height and undoubtedly regulating the local thickness of the cartilage. Combined with the observed introduction of the transverse clonal columns, oriented cell proliferation can provide fine surface generation and scaling (*Figure 4—figure supplement 2*). Moreover, the model highlighted the elegance of cartilage design involving transverse columnar clones in the sheet-shaped elements: this logic enables the uncoupling of thickness control (depends on cell numbers within a clone) and lateral expansion (depends on the number of initiated clones), which are likely two molecularly unrelated processes in vivo. The absence of a gradient during in silico simulations led to the generation of 3D asymmetrical clusters instead of straight columns (even in conditions of highly synchronized cell divisions, and starting from a laterally space-constrained initial configuration - suggesting the promotion of vertical growth due to space-exclusion in the lateral direction) (*Figure 5I,J*). This, in turn, led to the formation of surface irregularities in the cartilage with subsequent loss of local flatness (heat-map diagram in *Figure 5I,J*).

Importantly, lineage tracing also showed that for cartilaginous structures in the head with asymmetrical or complex irregular geometries, such as areas where several sheet-shaped cartilage elements were merged, clones were not constructed to perpendicular columns. In such locations, we identified irregular clonal clusters or randomly oriented clonal doublets, in accordance with the modelling results (*Figure 6A–J*). Thus, the shape and orientation of clones corresponds to the local geometry of the cartilage element.

Next, we attempted to target a molecular mechanism that controls the flatness and sheet-like shape of the facial cartilaginous sheets. We discovered that activation of ACVR1 (BMP type one receptor, ALK2) in developing cartilage leads to a phenotype with targeted clonal micro-geometries

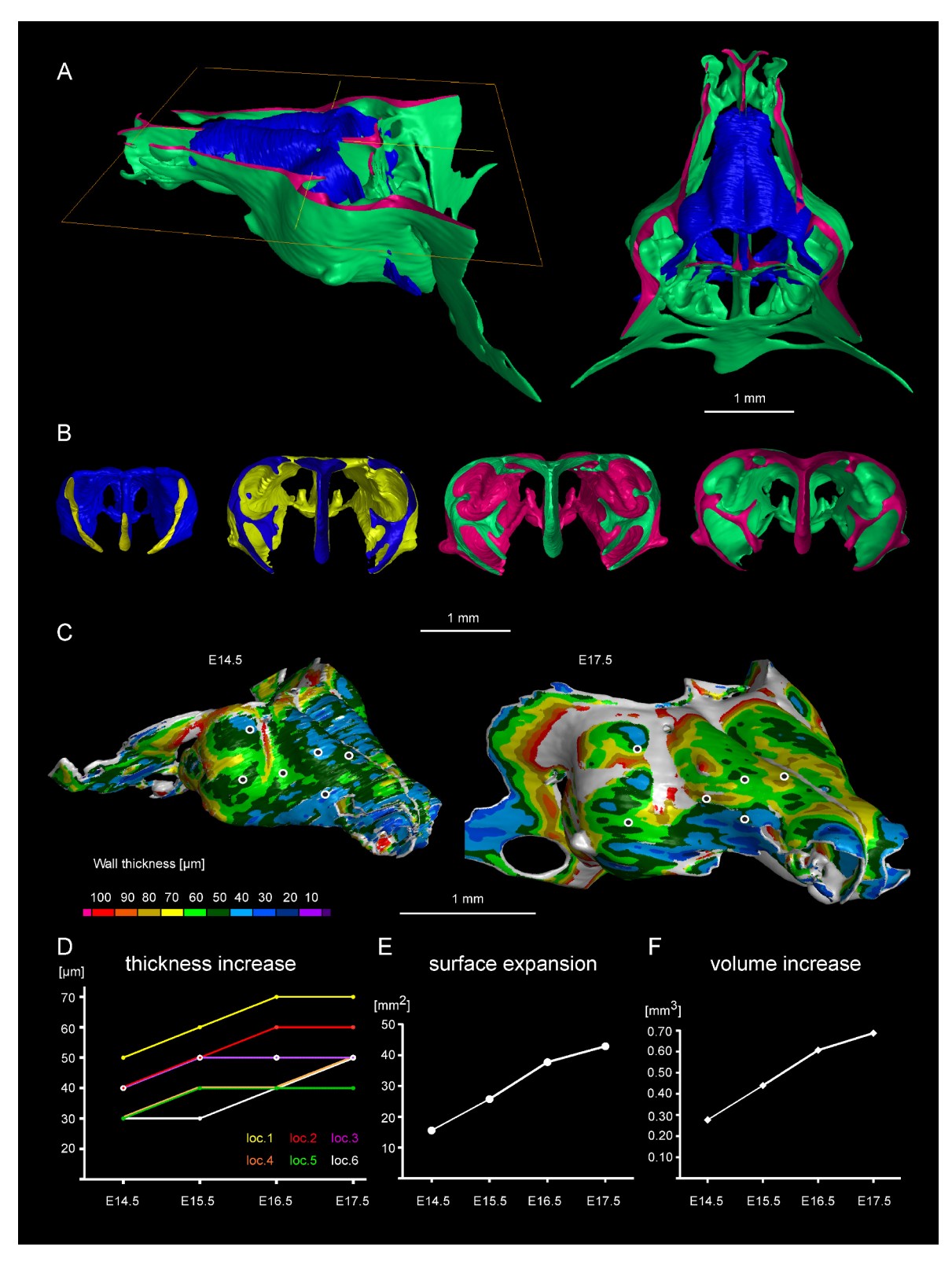

**Figure 3.** Facial chondrocranium undergoes major lateral expansion without extensive thickening during growth. (**A**) The 3D-model of E14.5 nasal capsule (blue) is placed onto the E17.5 model (green) for better presentation of growth-related changes. (**B**) Frontal clipping planes of 3D-models of nasal capsules at E14.5, E15.5, E16.5 and E17.5 (from left to right). Notice the mild changes in cartilage thickness as compared to the lateral expansion of the whole structure during growth. (**C**) Cartilage thickness heat-maps at E14.5 and E17.5 developmental stages. Less thick locations (color-coded in

*Figure 3 continued on next page*

*Figure 3 continued*

blue) correspond to intense growth zones shown in *Figure 11*. Dots show individual positions selected for precise measurements and demonstration on the graph shown in (D). Note that after E16.5 cartilage thickness remains relatively stable. (E–F) Cartilage surface area (E) and volume (F) expansion has been measured and compared between above mentioned stages. Note that there is a much greater increase in surface and volume (approximately 3-fold) than in thickness of the cartilage (less than 50%).

The following figure supplements are available for figure 3:

**Figure supplement 1.** 3D models and wall thickness analysis of chondrocraniums at different developmental stages.

**Figure supplement 2.** Comparisons of the shape and size differences between developmental stages and Wnt/PCP mutants.

---

(*Figure 6K–P,R*). We utilized a constitutively activated caALK2 transgene (*Fukuda et al., 2006*) together with genetic tracing in a way that every GFP-expressing cell is carrying constitutively active ACVR1. This experiment revealed a dramatic change of the shape of clonal envelopes, changing from straight perpendicular columns to disorganized spherical clusters inside the sheet-shaped cartilages of transgenic *Sox10-CreERT2/R26caALK2-IRES-GFP* embryos (*Figure 6K–N*). The ectopically activated ACVR1 resulted in the presence of clonal spherical clusters that interfered with the cartilage borders and caused the formation of ectopic bumps, swellings and other abnormal local shapes - in accordance with the mathematical modelling predictions (substantially resembling the condition with no gradient, see *Figure 5I*) (*Figure 6O–P*). All recombined cells in this caALK2 experiment became Sox9$^+$ chondrocytes. There were no other cell types found to be GFP$^+$, including perichondrial cells. This result indicates that BMP family ligands either produce the gradient that directs the orientated behavior of chondrocytes inside of the cartilage or, alternatively, that an experimental increase of BMP signaling renders the cells insensitive to the gradient established by other molecules. In any case, ACVR1 mutation can be used as a tool to change columnar arrangements into clusters (*Figure 6N,R*). The activation of ACVR1 by *Sox10-CreERT2* starting from E12.5 occurred both in perichondrial cells and in chondrocytes (based on our genetic tracing results using *Sox10-CreERT2/R26Confetti*). This later coincided with clonal bumps and bulging regions positioned mainly at the surface of sheet-shaped cartilaginous sheets (*Figure 6L,N,P*). These data also support the hypothesis that integration of clonal chondrocyte clusters into existing cartilaginous sheets likely depends on clonal shape and originates from the periphery of the cartilage. When this column-inserting process fails, the progeny of cells at the periphery of the cartilage forms ectopic bumps outside of the normal cartilage borders, and disrupts the flatness and straightness of cartilage surfaces.

Next, we attempted to block the planar cell polarity (PCP) pathway to challenge the system and disrupt the formation of perpendicular columns in the flat or curved cartilaginous sheets. To do this we performed μ-CT and EdU-incorporation analysis on Wnt/PCP mutants. Wnt/PCP pathway is well known for driving the cell and tissue polarity, and distinct facial phenotypes have appeared in Ror2, Vangl2 and Wnt5a homozygous mutants (*Figure 7A*). When EdU was administered 24 hr before embryo harvest, subsequent analysis showed no differences in the EdU-positive perpendicular clonal columns which formed within sheet-shaped facial cartilage of

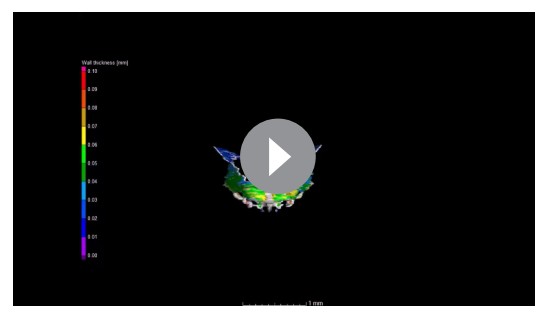

**Video 1.** 3D-models based on segmentation of mesenchymal condensations and mature cartilage from μ-CT tomographic data. The first sequence illustrates wall thickness analysis results represented as a heat-map, starting from E12.5 (facial mesenchymal condensation) until E17.5 (facial cartilage). Cartilages and other soft tissues shrink during contrasting with phosphotungstic acid, and, thus, the reported metrics cannot be directly compared with biological samples treated in a different way. The following sequence shows facial chondrocranium models of Wnt/PCP mutants in comparison to the wild type. The last sequence shows the full chondrocranium at different embryonic stages, followed by 3D models of both the control embryo and Wnt5a mutant embryo at E17.5.

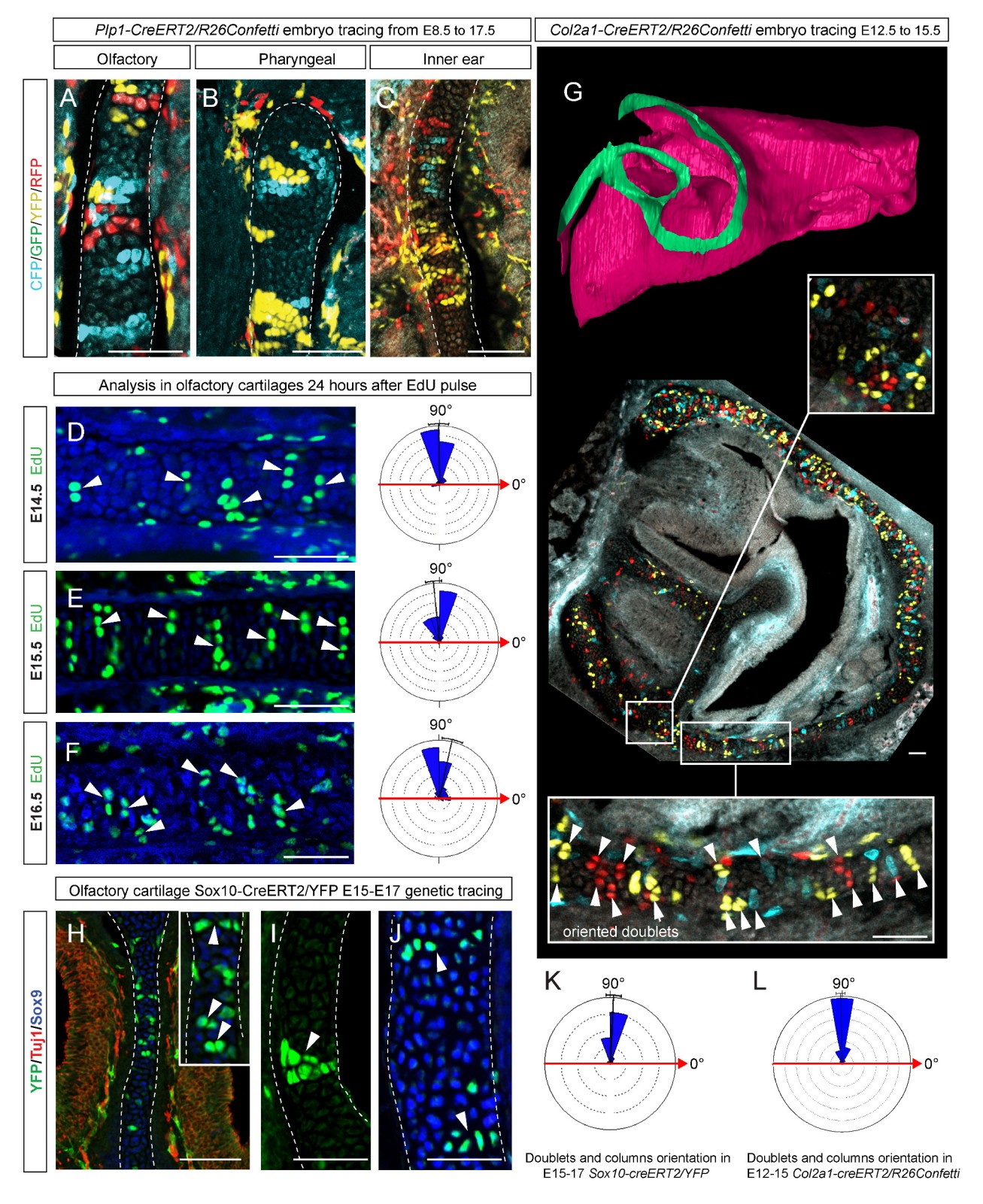

**Figure 4.** Clones of chondrocytes show transversely oriented columnar structure in sheet-shaped facial cartilage. (A–C) Chondrocyte clones at E17.5 were genetically traced from neural crest cells (E8.5), sagittal sections. The cartilage is outlined with white dashed line. (D–F) Analysis of EdU incorporation (24 hr after the pulse) into growing cartilage at different stages. Arrowheads indicate sparse columnar arrangements of EdU+ cells. Rose diagrams show orientation of EdU+ clusters in the cartilage of embryos at E14.5 (D), E15.5 (E) and E16.5 (F). (G) Genetic tracing of chondrocytes

*Figure 4 continued on next page*

*Figure 4 continued*

initiated at 12.5 and analyzed at 15.5. The clipping plane of a 3D-model (side projection) is shown for better orientation in the analyzed region. Note the transverse orientation of clonal doublets and columns (arrowheads). (**H–J**) Genetic tracing induced at E15.5 and analyzed two days later in embryos of *Sox10-CreERT2/R26YFP* mouse strain. Arrowheads indicate clonal columns of chondrocytes that formed inside of the growing cartilage between E15.5 and E17.5. The orientations of clonal arrangements are shown in the rose diagram in (**K**). (**L**) Orientation of clonal doublets and columns in genetically traced cartilage (from E12 to E15) of *Col2a1-CreERT2/R26Confetti* embryos. Scale bars = 100 µm.

The following figure supplements are available for figure 4:

**Figure supplement 1.** Oriented clonal dynamics in chondrogenic mesenchymal condensations.

**Figure supplement 2.** Clonal oriented clusters of chondrocytes contain closely associated perichondrial cell in flat facial cartilages.

Wnt5a knockout mutants or wild type controls (*Figure 7B–D*). µ-CT analysis at early developmental stages showed that as early as at E12.5, Wnt5a mutants had abnormal shape and placement of the mesenchymal condensations that create a template for future cartilaginous structures (*Figure 7E–F*). Although µ-CT analysis of Wnt5a, Ror2 and Vangl2 homozygous mutants at later developmental stages confirmed that chondrocranium shape was heavily affected (with generally shortened nasal capsules as compared to both wild-type and heterozygous controls) (*Figure 7A*), we did not detect any defects in cartilage micro-geometry, including thickness or surface organization. Altogether, these results indicate that Wnt5a, Ror2 and Vangl2 do not control cartilage growth and shaping per se (via the insertion of perpendicular columns). Instead, they influence the position and shape of chondrogenic condensations, which define the future geometry of the facial chondrocranium (*Figure 7E,F*).

Following the prediction from our mathematical model that the thickness of the cartilage can be controlled by the number of cells in the inserted clonal column, we searched for the molecular mechanisms which control this. Knowing from our results that proliferation rate drops in the mature cartilage, we hypothesized that chondrocyte maturation speed may influence the number of cell divisions within a column. To test this suggestion, we analyzed G-protein stimulatory α-subunit (Gsα) knockout embryos (*Figure 8*). Inactivation of Gsα, encoded by *Gnas,* is known to lead to accelerated differentiation of columnar chondrocytes, without affecting other aspects of cartilage biology (*Chagin et al., 2014*). We analyzed three different locations in the developing chondrocrania, and observed a significant reduction of cartilage thickness in absolute metrics (*Figure 8A,B,J*), as well as in terms of the number of cells within each column (*Figure 8B–H*). Thus, the Gsα knockout is a perfect tool to test whether the modulation of differentiation speed can be used to create a variation of local cartilage thickness. The result of this experiment demonstrated that sheet-shaped cartilages in Gsα knockout embryos are thinner than that of littermate controls, while other parameters (including general size and shape of nasal capsule and other locations in the head together with the transverse orientation of chondrocyte columns) remain largely unaffected (*Figure 8A,B,I*). Thus, these data experimentally validated mathematical predictions and confirmed that the thickness of cartilage is determined by the number of cell divisions within a transverse clone, and that this is uncoupled from lateral expansion.

Next, we wanted to know how clonal cell dynamics accounts for the shape development in rod-shaped cartilages. For this we investigated the clonal dynamics in Meckel, rib and limb cartilages with the help of Confetti-based genetic tracing as well as EdU incorporation. The clonal arrangements appeared highly oriented and strongly resembled the clonal columns we observed in the facial cartilage. The columnar clones were oriented mostly transversally in the plane of a rod diameter and could not explain the early growth along the main axis of the skeletal element (*Figure 9*). These tracing results suggested that longitudinal extension is based on continuous development of chondrogenic mesenchymal condensations on the distal tip and is followed by the transverse proliferation of chondrocytes, which accounts for the proper diameter of a cartilaginous rod. The logic of oriented cell dynamics in sheet-shaped and rod-shaped cartilages is summarized in *Figure 10*.

Since the integration of clonal units is likely to be uneven in the cartilage, we questioned how the anisotropy of local proliferation can impact the shaping processes on a macro-scale. Starting from E14.5, the olfactory capsule is already formed of mature chondrocytes. Indeed, in this structure,

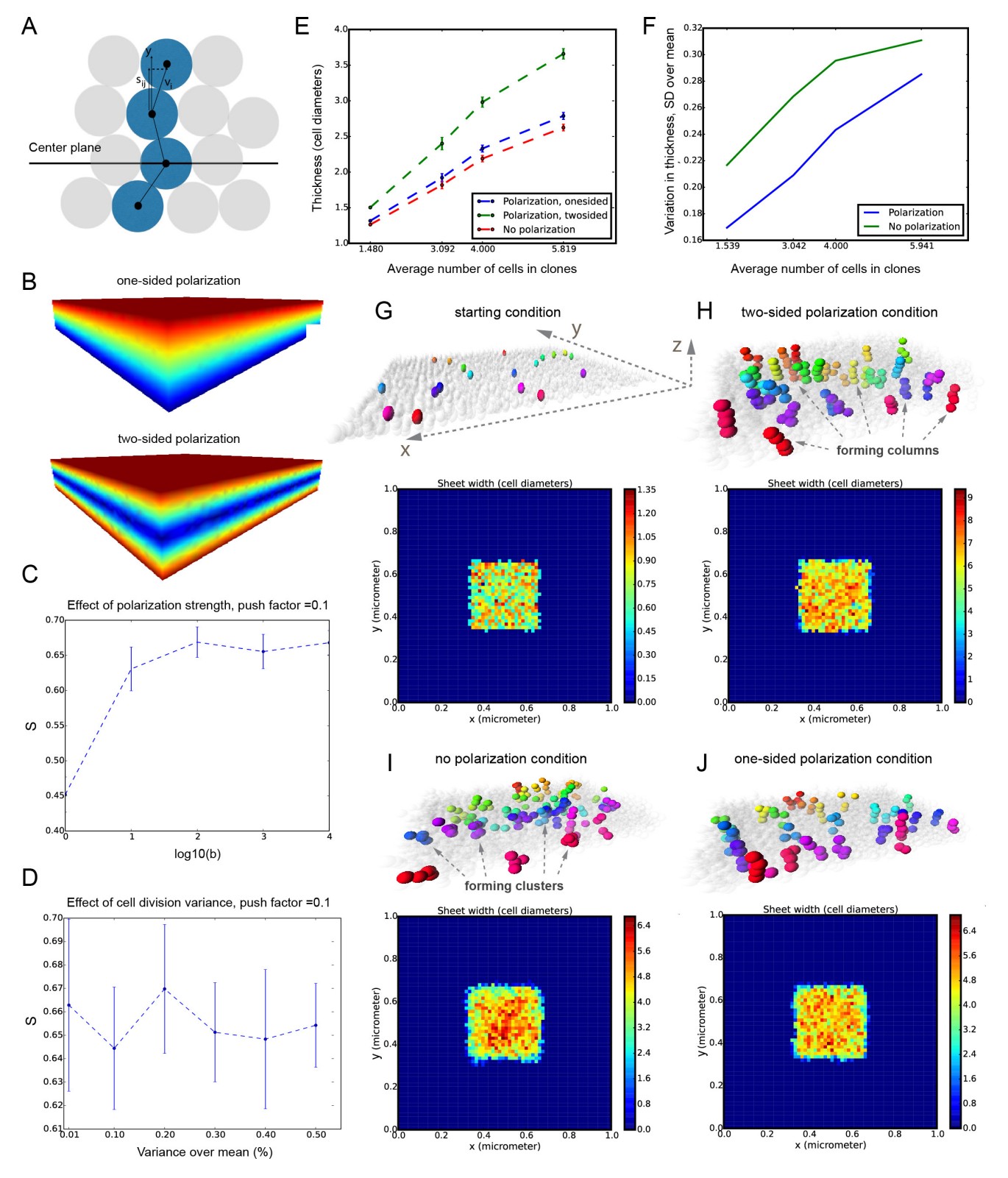

**Figure 5.** Mathematical model of cell dynamics during sheet-shaped cartilage development and growth. (**A**) Transversal (along z-axis) clipping plane showing conceptual arrangements of modelled cells within the layer as a result of a typical simulation. The degree of microstructure order, S, is measured by the sum of orthogonal projections on the unit vector in the z-direction, normalized by the number of cells. (**B**) Visualized and modelled one- and two-sided gradients used to direct oppositional growth of the clonal columns during computer simulations. (**C**) The degree of determinacy in

*Figure 5 continued on next page*

*Figure 5 continued*

the response to the external gradient is modeled by a parameter, *b*, where a high value results in near perfectly polarized cell divisions (pushing may still introduce randomness in the eventually chosen site) and where the limit *b* tends to zero results in completely random division directions. As can be seen, the degree of microstructure order, and hence columnar growth, increases with the strength of the polarization response. (**D**) For a strongly polarized cell, the model predicts that even a large variation in the individual cell division times results in only a moderate decrease in the columnar order. (**E**) Graph showing the dependence of cartilage thickness on the absence or presence of one- and two-sided polarizing gradients. (**F**) Graph showing how the regularity of the thickness depends on the presence of a polarizing gradient. Note that, based on (**E–F**), the conditions with polarization demonstrate higher regularity and thickness over multiple locations. (**G–I**) Snapshots of typical in silico simulations of cell dynamics during sheet-shaped cartilage development: layers of chondrogenic cells demonstrated in 3D before (**G**) or after simulations (**H–J**) shown together with 2D heatmap diagrams of cell layer thickness irregularity (below) represented as a view from above (x,y dimensions). Clonal progeny is represented as individually color-coded cellular clusters or columns in 3D visualizations. Note the high degree of thickness irregularity that corresponds to the variety of differently oriented clonal shapes in condition with no polarizing gradient (**I**). The highest geometrical regularity of the modelled cartilaginous sheet together with stereotypical columnar clonal arrangement is achieved in the condition with two-sided polarizing gradient (**H**).

proliferation was localized to specific regions, but remained generally low elsewhere (*Figure 11A–B*) according to the analysis of EdU incorporation. As we demonstrated above, proliferative regions expand due to the active integration of new clonal columns and clusters. We projected the low- and high proliferative zones onto the 3D structure of the nasal capsule at E13.5-E15.5 to understand not only the dynamics of lateral expansion, but also to see how the local expansion of cartilage may influence bending and geometrical changes on a large scale (*Figure 11C–F*). Since proliferative zones in nasal capsule are restricted and have defined edges, they inevitably induce tension and bending of the surrounding cartilage sheet.

In order to address the logic of distributed proliferative zones and its role in shape transitions between stages we took advantage of the mathematical model developed by the Enrico Coen and Andrew Bangham laboratories. This model has been efficiently validated and applied for advanced simulations of complex 4D plant organ development (*Green et al., 2010*; *Kennaway et al., 2011*). To simulate in silico nasal capsule shape transition from E13.5 to E14.5, we generated a basic E13.5-like shape by converting a sheet-shaped growing trapezoid into a corresponding 3D structure (*Figure 12A*, central part and *Video 2*). The result was considered as a simplified starting condition for further simulations. Next, two lateral zones with a low rate of proliferation were introduced according to their original position in E13.5 nasal capsule. Further simulations of the growth showed that these low proliferative zones impose a characteristic bending on the sides of the simulated structure. This bending corresponds to the lateral transformations observed in embryonic development of the nasal capsule between E13.5 and E14.5 (*Figure 12B–C*). This characteristic lateral bending did not depend on anterio-posterior polarity in the cartilage or formation of the groove at the midline (*Figure 12D*). According to the model, the polarity only affected the potential for the anterior elongation due to the anisotropic growth of the entire cartilaginous structure. Our results also suggested that the nasal septum functions as a slower proliferating anchoring point to the roof of the nasal capsule, which is necessary for the formation of the midline groove at E14.5. A simulated groove at the midline provided for the general bend and flattened shape of the in silico cartilage, similar to the native E14.5 nasal capsule and contrary to the model without the simulated midline groove (*Figure 12C–D*).

To validate the general rules of in silico transformations, we performed material modelling using plastic film to simulate anisotropic expansion and bending due to integration of local growing zones with attached borders. This simple material modeling demonstrated that growth zones/local expansions in the flat planes generate mechanical tensions which bend the structure (*Figure 12E*). We then performed another material modelling experiment using isotropic thermal expansion/constriction of a plastic film. For this purpose, we drew black regions (analogous to the lateral low proliferative zones in E13.5 nasal capsule) onto white plastic film that was cut in a shape of a trapezoid capable of transforming into a nasal capsule-like dome. Under the heating provided by a thermal infrared lamp, the black zones received more heat and isotropically shrunk. Shrinkage of the black zones created physical tensions that eventually bent the structure in a way similar to the original nasal capsule geometry at E14.5 (*Figure 12F*). The model with shrinking zones is comparable to the real growth conditions as the nasal capsule expands faster than spatially distributed slow

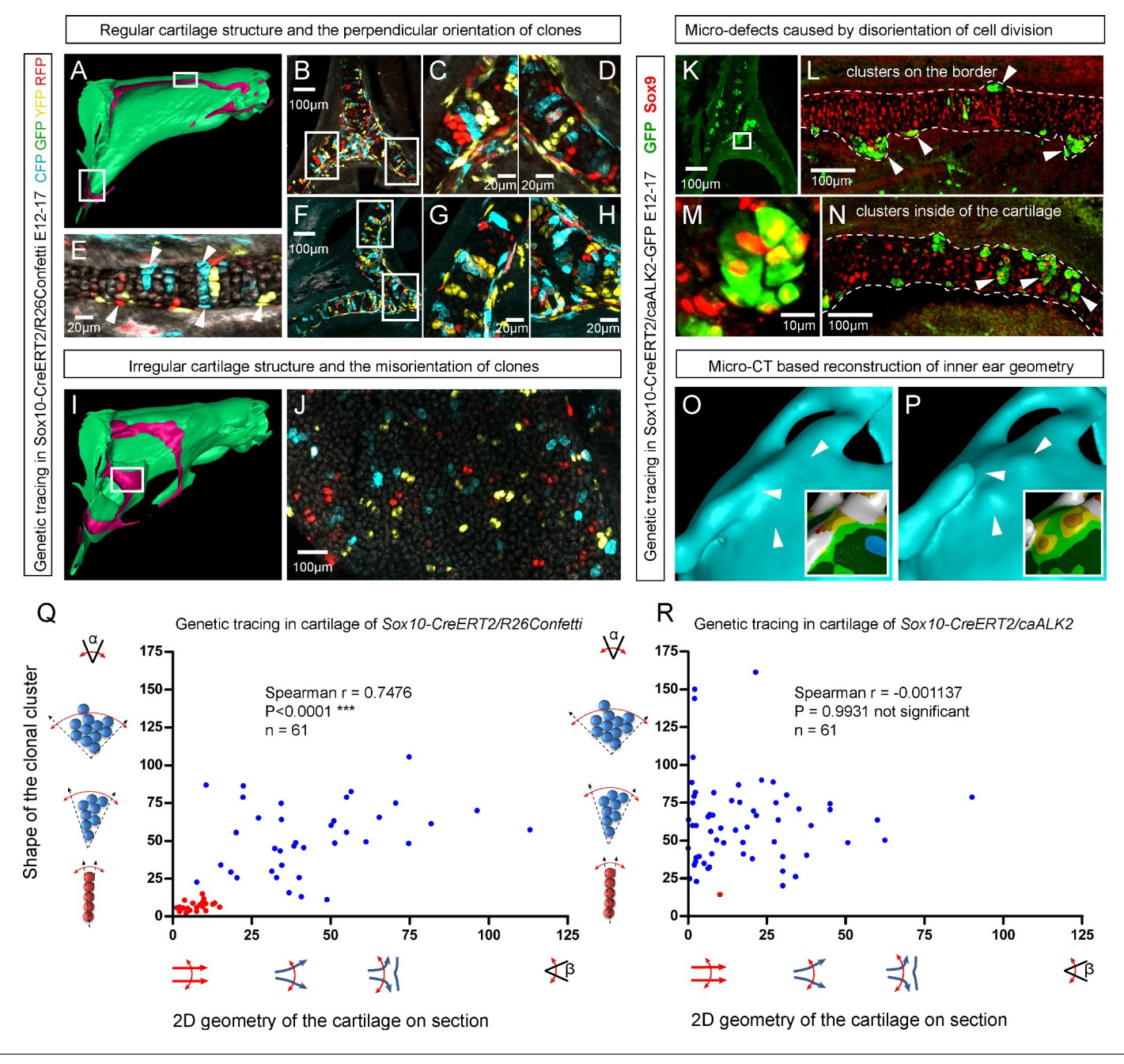

**Figure 6.** Clonal arrangements of chondrocytes influence local geometry and cartilage surface. (A–H) Columnar clonal arrangements in sheet-shaped cartilages of facial chondrocranium visualized with genetic tracing in *Sox10-CreERT2/R26Confetti* embryos. (A) 3D-model with a clipping plane shown as a side projection. White frames show locations analyzed in (B–H). Panels (C,D) and (G,H) represent magnified areas outlined in (B) and (F), respectively. (E) Traced perichondrial cells at the base of chondrocyte columns that share a clonal origin and are indicated by arrowheads. Note that rich tracing in the perichondrium correlates with highly efficient tracing in the cartilage (compare, for example, (B–D and E). (I–J) Clonal clusters show no columnar structure in geometrically irregular elements such as junctions and fusion points of several cartilaginous elements (highlighted in 3D-model with frame). (K–P) Cre-based activation of ACVR1 in facial sheet-shaped cartilage elements of *Sox10-CreERT2/stop^{floxed/floxed}caAlk2-IRES-GFP* embryos induced at E12.5 and analyzed at E17.5. Locations are the same as highlighted in (A). (K–N) Green clusters are sparse and clonal and show successful activation of ACVR1. Note the formation of spherical clusters of chondrocytes instead of transversely oriented columns. Spherical clusters bulging from the sheet-shaped cartilage are indicated by arrowheads in (L). Amorphous clusters caught inside of the structure are indicated by arrowheads in (N). The cartilage surface is outlined with a dotted line. (O–P) Despite low efficiency of Cre-based ACVR1 activation, the local disruptions of cartilage 3D geometry (analyzed with μ-CT) take place: the inner ear capsule is affected by bulges and the connecting junction is destroyed as indicated by arrowheads. Thickness heatmaps of analyzed location show local thickening of the cartilage as a result of non-oriented placement of chondrocytes with disrupted

*Figure 6 continued on next page*

*Figure 6 continued*

BMP signaling. (**Q–R**) Graphs showing how the regularity of the cartilage (flatness) correlates with orientation of clonal envelopes in the cartilage of *Sox10-CreERT2/R26Confetti* (**Q**) and *Sox10-CreERT2/stop^{floxed/floxed}caAlk2-IRES-GFP* (**R**) embryos. The angle α characterizes the elongation of a clonal cluster consisting of multiple cells, as shown in a legend of a corresponding graph axis. Small values of α correspond to highly oriented clonal envelopes such as vertical clonal columns. Angle β is the angle between two opposite cartilage surfaces framing cartilage tissue in this locality. Sheet-shaped cartilages have almost parallel surfaces and angle β values are normally set between zero and 20 degrees. Note that the population of clonal columns (red dots) is almost completely eradicated from the cartilage when ACVR1 is activated in (**R**).

proliferative regions (simulated as shrinking zones inside of the non-expanding plastic material). These results, combined with analysis of proliferation and 3D visualizations, strongly suggest that the distribution of uneven proliferative zones plays an important role in the shaping of the facial chondrocranium during embryonic development.

Taken together, we reveal a set of principles contributing to the accurate scaling and shaping of cartilage tissue during growth. The reverse engineering of this process highlights the involvement of highly specialized systems that control the directional growth at the levels of micro- (clonal shapes) and macro-geometries (proliferative regions in nasal capsule). Our results show that allometric growth of complex 3D cartilage elements is not achieved by simple, evenly distributed and/or unidirectional proliferation, but is sculpted by precisely localized proliferation.

## Discussion

Here we report the discovery of how oriented cell behavior and molecular signals control cartilage growth and shaping. Previously, the use of chimeric avian embryos demonstrated the competence of facial mesenchyme in producing species-specific shapes and sizes of cartilage elements (*Eames and Schneider, 2008*), while facial epithelium and brain provided the instructive signals guiding generalized shaping of the face (*Chong et al., 2012*; *Foppiano et al., 2007*; *Hu et al., 2015*). Knowledge of how the facial cartilaginous elements are shaped has been rather restricted, and mainly concerned with the correct formation of chondrogenic mesenchymal condensations.

The accurate expansion of the chondrogenic condensation or cartilage during growth is no trivial matter. The general shape should be both preserved and modified at the same time. We show that anisotropic proliferation and oriented clonal cell dynamics are implemented to achieve the necessary outcome. The reverse engineering of this process highlighted the involvement of highly specialized systems that control the directional growth at the levels of micro- (clonal shapes) and macrogeometries (proliferative regions).

Allometric growth of complex 3D structures requires certain cellular logics and cannot efficiently proceed with equally distributed and/or unidirectional proliferation inside of the mesenchymal condensation or cartilage element. On the other hand, we did not observe the formation of growth plate-like zones in early sheet-shaped (nasal capsule) or rod-shaped cartilages, nor uniform expansion of cartilage in all directions. Thus, the underlying growth and shaping mechanisms required an explanation.

To test various strategies of cellular behavior during cartilage growth we devised a model simulating different aspects of multicellular dynamics in 3D together with lineage tracing of individual clones. Most of the currently existing models of cell dynamics and tracing operate in 2D space, which often limits the predictions (*Jarjour et al., 2014*). Our model suggested that a gradient-controlled orientation of clonal expansion can explain the biological observations (i.e. it is consistent with ordered columnar growth and its disruption results in spherical microdomains rather than columns), and showed the relation between the geometries of clonal domains (envelopes), the overall shape and the fineness of the surface. We confirmed the predictions from the model in a series of experiments involving tracing with multicolor reporters and manipulating the cartilage with mutations. Our results showed that the formation of oriented clones of chondrocytes with clonal envelope shape corresponds to the geometry of the analyzed locality. The sheet-shaped cartilage elements consisted of transversely oriented clonal columns, while asymmetric complex geometries revealed a variety of clonal shapes ranging from spherical to particularly oriented.

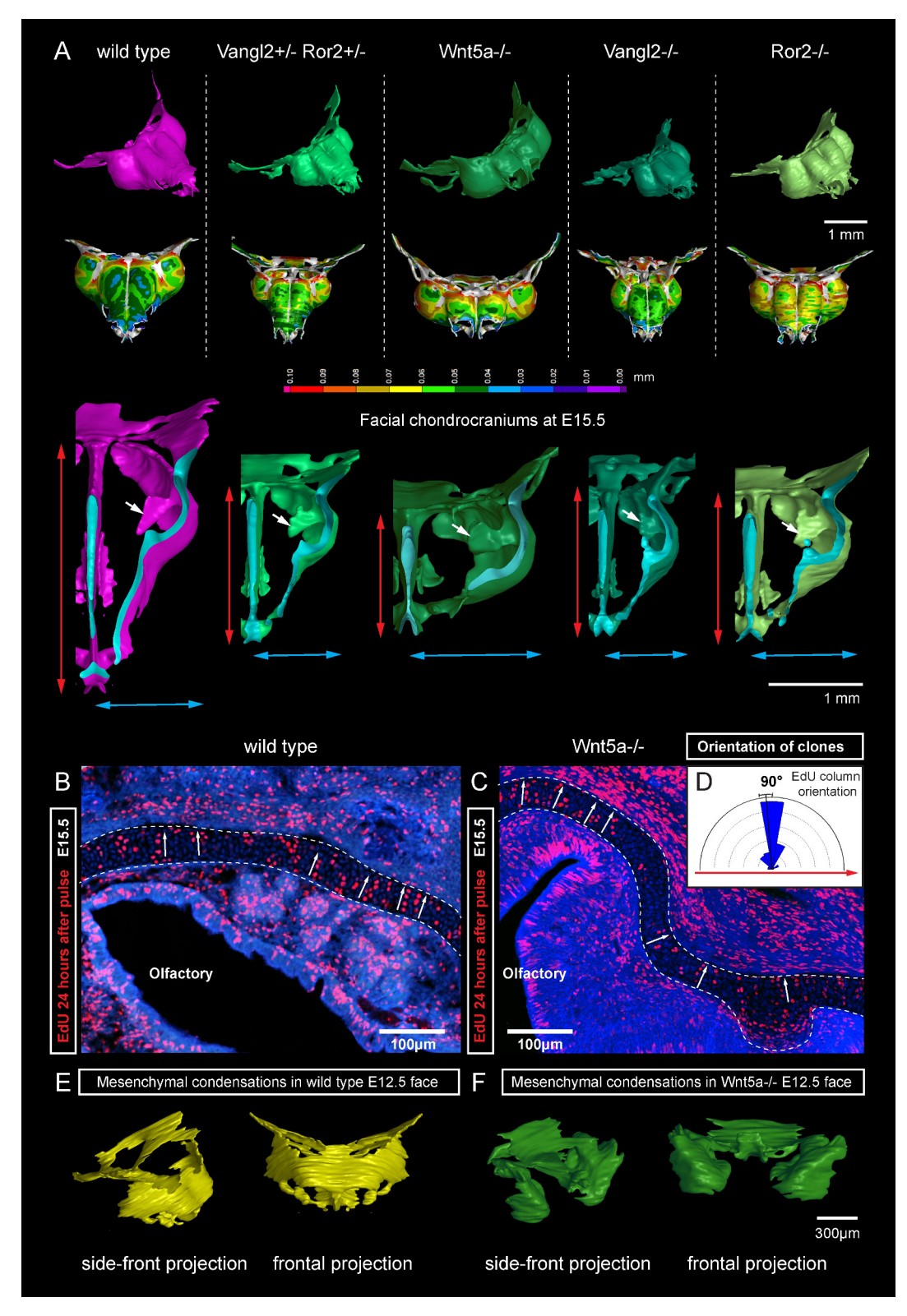

**Figure 7.** Deficiencies in Wnt/PCP pathway reshape the chondrocranium. (**A**) μ-CT-based reconstructions of the facial chondrocranium of wild type control, *Wnt5a$^{-/-}$*, *Vangl2$^{-/-}$*, *Ror2$^{-/-}$* and *Vangl2$^{-/+}$/Ror2$^{-/+}$* embryos at E15.5, with wall-thickness analysis (the row below). Clipping planes in the top projections show that all major- and fine structures (indicated by white arrows) are in place in the Wnt/PCP mutants. Red and blue arrows help to compare the width and the length of the chondrocranium. (**B–D**) Analysis of EdU incorporation in the facial sheet-shaped cartilage, 24 hr after the

*Figure 7 continued on next page*

*Figure 7 continued*

pulse: control (**B**) and *Wnt5a⁻/⁻* mutant (**C**) embryos. Sparse clusters and columns of EdU⁺ cells correspond to clonal arrangements previously shown with genetic tracing in the same locations. Note that the orientation of chondrocyte placement in the cartilage is not affected in the mutant embryos. Quantification is presented in a rose diagram in (**D**). For control, we refer to the rose diagrams in *Figure 3*. (**E, F**) μ-CT-based 3D reconstruction of mesenchymal condensations at the developmental stage E12.5 in control (**E**) and *Wnt5a⁻/⁻* mutant (**F**) embryos shows their misplacement in a mutant.

Genetic tracing initiated during transition of condensations into cartilage resulted in clonal columns within both sheet- and rod-shaped cartilage elements. This confirms that chondrogenic condensations undergo complex oriented cell dynamics during their development. Importantly, tracing of chondrocranium cartilage showed formation of transverse clonal columns as growth proceeded. Intercalation of newly born columns into pre-existing cartilage provided for the expansion potential in the sheet-shaped cartilage. This growth mechanism is very original and is not reported elsewhere so far.

A few studies have demonstrated how clonal envelopes form in accordance with the general shape of the structure. These were mainly conducted on *Drosophila* imaginal wing disc or growing flower petals. In all cases the authors highlighted that the shape of clonal geometries correlates with the major vector of expansion in the growing structure (*Green et al., 2010*; *Repiso et al., 2013*; *Strutt, 2005*). This implies the presence of polarized activity that directs the shaping of the tissue. Here, we provided the first experimental evidence of how the control of the directional clonal expansion influences the shape of a vertebrate tissue on a large scale. Moreover, in the sheet-shaped cartilage the orientation of clonal domains, i.e. the columns, does not correspond to the vectors of major expansion, but rather serves for uncoupling lateral expansion control and thickness tuning. In line with that, the number of chondrocytes comprising the clonal column or cluster depends on Gsα-mediated signals. Variations in this number do not significantly affect the lateral dimensions of the whole sheet-shaped cartilage structure: the thickness of the cartilage becomes less while the general geometry and size stay preserved. Additionally, the shape and orientation of clonal envelopes in cartilage is partially controlled by BMP signaling, since micro-geometries of clones depend on activation of ACVR1. Based on these results, we assume that BMP ligands (because of cAlk2/ACVR1 phenotype affected clonal orientation) expressed around the regularly shaped cartilages may play a role similar to the in silico predicted gradients. Indeed, the expression of INHBA, BMP5 and BMP3 fit this expression profile quite well (according to Allen Developing Mouse Brain Atlas (http://developingmouse.brain-map.org) and Eurexpress (http://www.eurexpress.org) in situ public databases). At least, BMP5 is clearly expressed at the cartilage periphery and has been shown to affect the cartilage shape by David Kingsley lab (*Guenther et al., 2008*).

Our experimental manipulations of planar cell polarity (PCP) pathway did not affect microgeometries and clonal domains, but strongly affected the chondrocranium shape on the macroscopical scale in several different ways. These phenotypes appeared to be rooted in pre-chondrogenic or early chondrogenic stages, and are based on distorted placement of mesenchymal condensations in the very early head. These experiments with Wnt/PCP mutants may potentially provide a better understanding of species-specific mechanisms of control and evolution of the facial shape on a macro scale.

Regular shapes require regular cellular arrangements and clonal cell dynamics. It is not only sheet-shaped cartilage in the head that demonstrate geometric regularity; rod-shaped cartilage (Meckel, embryonic ribs and long cartilages in limbs) also has a regular shape. Regular clonal patterns, conceptually similar to those found in sheet-shaped cartilage, explain conservative tissue dynamics during formation and growth of cartilaginous rods. Indeed, genetic tracing experiments suggested that formation of clonal columns is important for the diameter control, while chondrogenic condensations at the very tip of the rod-shaped growing structures enable elongation. Similar to the cell dynamics in the sheet-shaped cartilage, this mechanism may provide for uncoupling of length versus diameter control. Such uncoupling may generally enable developmental and evolutional plasticity of cartilage size and shape.

The mechanism controlling the thickness or diameter of sheet-shaped and rod-shaped cartilage elements not only includes spatially orientated behavior, but also involves the regulation of cell number within each chondrogenic clone. Immature chondrocytes are proliferatively active, while more

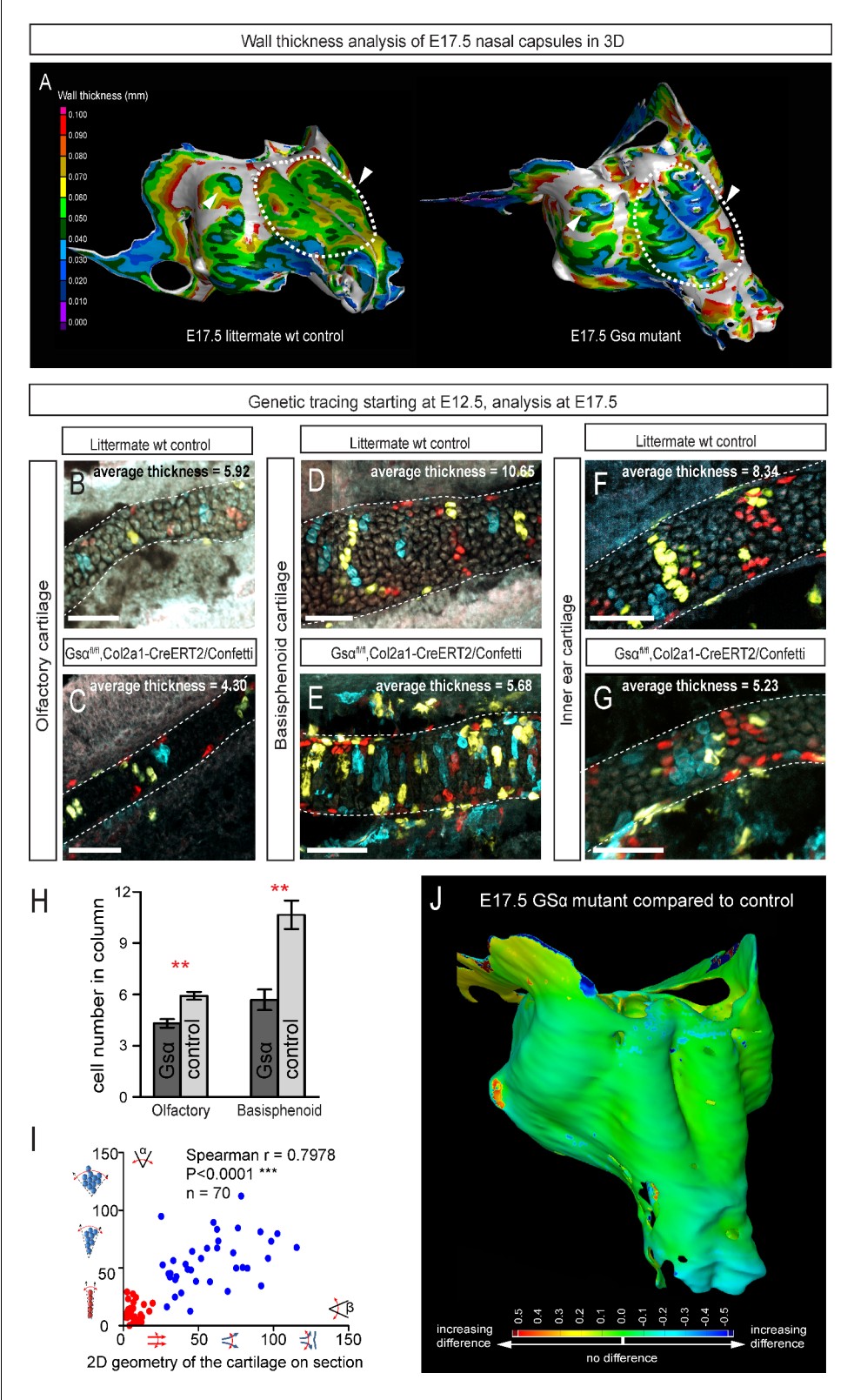

**Figure 8.** Manipulation of cartilage thickness in GSα mutant embryos. (**A**) Wall thickness was analysed in the μ-CT segmented olfactory system of control (left) and GSα (G-protein subunit alpha) mutant (right). Large areas with decreased cartilage thickness are highlighted with a dashed line and white arrows. (**B–G**) Clonal genetic tracing of chondrocyte progenitors and chondrocytes induced by tamoxifen injection at E12.5 in *Col2a1-CreERT2/ R26Confetti/GSα* ^floxed/floxed^embryos (**C,E,G**) and littermate controls (**B,D,F**) at E17.5. (**H**) Quantification of cartilage thickness in the olfactory system and
*Figure 8 continued on next page*

*Figure 8 continued*
basisphenoid from three independent experiments. Note the significant decrease of cartilage thickness in all analyzed locations. Oriented organization of the chondrocyte clones was not affected by GSα ablation. The difference between control (mean = 5.9, sem = ±0.23, n = 4) and mutant (mean = 4.3, sem = ±0.25, n = 3) olfactory cartilage thickness is significant (p=0.0053). The difference between control (mean = 10.6, sem = ±0.83, n = 3) and mutant (mean = 5.7, sem = ±0.61, n = 3) basisphenoid cartilage thickness is significant (p=0.0087). Scale bars = 100 µm. (I) Graph showing that the regularity (straightness) of the cartilage correlates with the orientation of chondrocyte clones in the cartilage of *Col2a1-CreERT2/R26Confetti/GSα* [floxed/floxed] embryos. Angle α characterizes the orientation of chondrocyte clones consisting of multiple cells as shown in a legend of a corresponding graph axis (y). Small values of α correspond to highly oriented chondrocyte clones such as transverse clonal columns. Angle β is the angle between two opposite cartilage surfaces. Since sheet-shaped cartilage elements have almost parallel surfaces the angle β was normally set between zero and 20 degrees. (J) GOM Inspect software was used to compare the shape of the nasal capsule between GSα mutant and control embryo at E17.5.

mature chondrocytes show decreased proliferation. Therefore, differentiation speed emerges as a concept which could regulate the organ shape by impinging on clone size, thereby altering the thickness or diameter of the cartilage. This concept is known to operate in the brain and other tissues with classical stem cell/transiently amplifying cell arrangements (*Díaz-Flores et al., 2006*).

Clonal genetic tracing and EdU labeling experiments suggested that the origin of clonal columns and clusters might be represented by the cells located at the periphery of forming cartilage. The spherical clusters of chondrocytes forming at the periphery of the cartilage in cALK2 mutant mice may suggest that the cell source is also located at the periphery and might be a perichondrial cell. Clonal relationships between perichondrial cells and columns of chondrocytes also support the hypothesis of perichondrial cells acting as a stem population during cartilage expansion. In general, the heterogeneity and multipotency of perichondrial cells is still unclear, although there are multiple studies showing the perichondrium as a source of chondrocytes and osteoblasts (*Kobayashi et al., 2011*; *Li et al., 2017*; *Maes et al., 2010*).

In addition to this, the perichondrium might mediate non-autonomous effects in the cartilage in case of cAlk2 and GSα experiments. Genetic tracing shows that some perichondrial cells always recombine with *Sox10-*, *Plp1-* and *Col2a1-CreERT2* lines, and, in case of functional experiments, may indirectly control some evens in more mature layers. Also, it is not clear how the fine border of the cartilage is set, and whether the perichondrial layer may play a key border-setting role during development and regeneration. This should be investigated further.

Next, our results show that tuning of macro-geometries on a large scale can be achieved through a stage-specific placement of proliferative hot zones where new clonal domains intercalate into the main cartilage structure. Anisotropic heterogeneous proliferation is a powerful tool, which, together with polarity in the tissue and local patterning, can drive the organ shape development (*Ben Amar and Jia, 2013*; *Campinho and Heisenberg, 2013*). The localized growth zones provide for the general expansion and also bend the cartilage by creating local tensions that require mechanical relaxation and influence further development of the overall shape (*Schötz et al., 2013*). For probing such transformations of the sheet-shaped facial cartilage we applied an in silico model that was already successfully validated in a number of growth, shaping and scaling tasks (*Green et al., 2010*; *Kennaway et al., 2011*). Such a model was necessary to understand why the high and low proliferation zones are positioned in such a specific way. Indeed, the discovered distribution of proliferative zones in the whole nasal capsule did not help us per se with intuitive explanations of geometrical changes on the macro-scale. Despite this counter-intuitive dataset, the mathematical model provided an insight into the logic of the high and low proliferation zones in relation to a transition between investigated cartilage shapes.

For example, it turned out that the position of lateral slow proliferation zones enables the generation of the symmetrical bends at the sides of the nasal capsule during transition from E13.5 to E14.5 developmental shapes. Furthermore, real material modelling confirmed the results predicted by the mathematical model, and generated lateral bends similarly to the native structure. The molecular mechanism controlling the dynamic distribution (patterning) of fast/slow proliferative zones in the cartilage is still unknown. It is likely linked to developmental signals from other tissues such as the olfactory epithelium or the mesenchyme surrounding the cartilage. Identification and validation of these signals will be essential in future studies and would involve a substantial combination of screening and functional approaches with transgenic animal models.

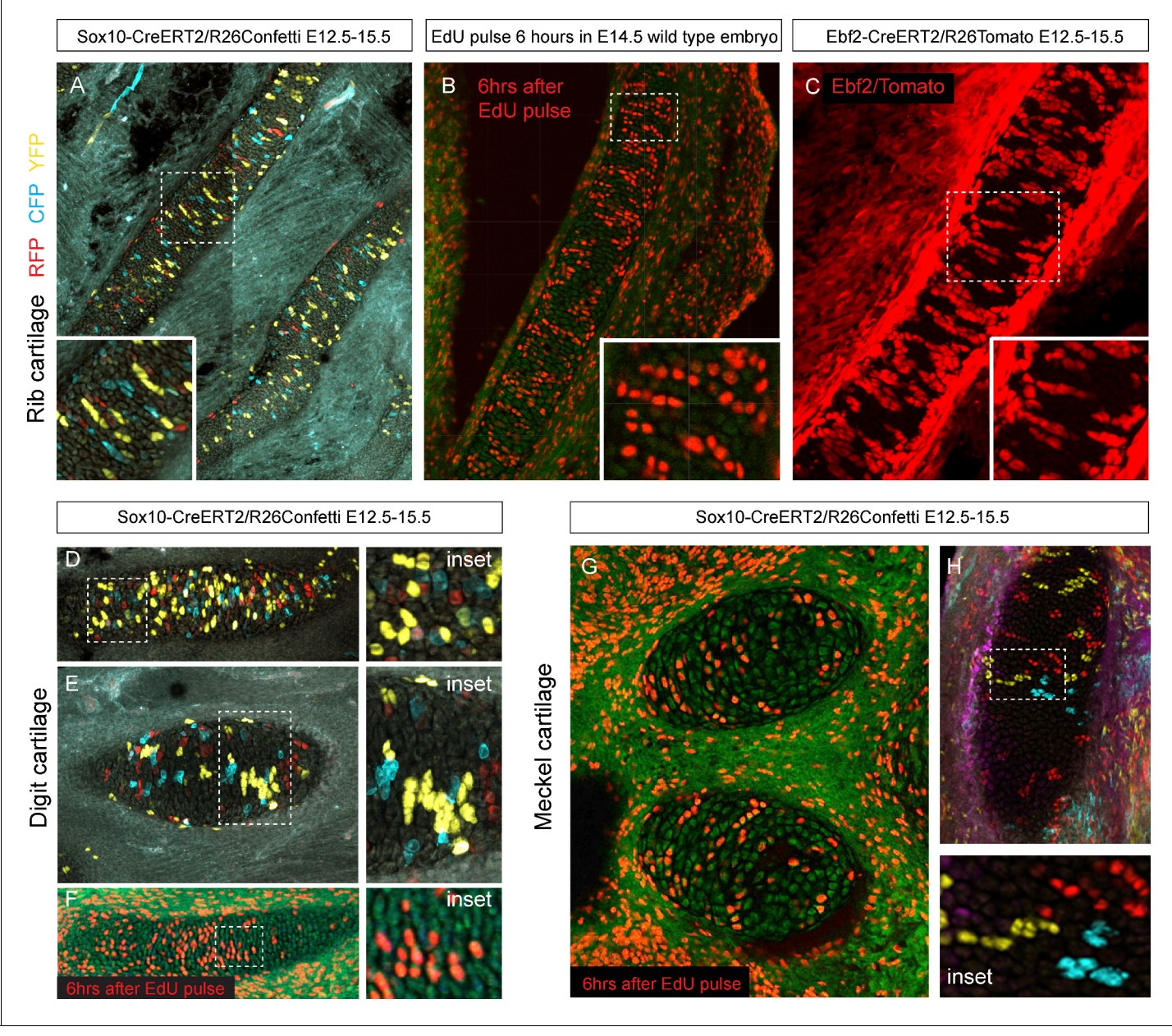

**Figure 9.** Oriented cell dynamics during development of rod-shaped cartilage elements. (**A**) Genetic tracing in developing rib cartilage. Note the transverse pattern of chondrocyte clones. Dotted rectangle shows the area of magnified inset on bottom left. (**B**) 6 hr after EdU pulse in E14.5 embryo, transverse patterns were observed in ribs. Dotted rectangle shows the areas of magnified inset in bottom right corner. (**C**) Genetic tracing in developing rib cartilage shows transverse patterns. Dotted line represents area magnified in inset on the bottom right. (**D–F**) Genetic tracing (**D,E**) and 6 hr after EdU pulse (**F**) in developing digit cartilage of the upper limb. The areas of magnified insets (located on the right side) are outlined by dotted lines. (**G**) Transverse patterns in developing Meckel cartilage resulting from EdU pulse and analysis 6 hr after administration. (**H**) Genetic tracing shows transverse orientation of clonal chondrocytic columns in the Meckel cartilage. Dotted line shows the area magnified in the inset on the right.

The anisotropic proliferation can be an important evolutionary mechanism that is directly responsible for the differences in snout geometry in a variety of phylogenetic groups. Additionally, it might be important for understanding the development of the facial shape variation in humans (*Sheehan and Nachman, 2014*) as well as numerous pathologies (*Afsharpaiman et al., 2013*).

One alternative way to fine-tune macro-geometry of a cartilage element is to continuously add on pre-shaped chondrogenic mesenchymal condensations from the pool of competent progenitors that

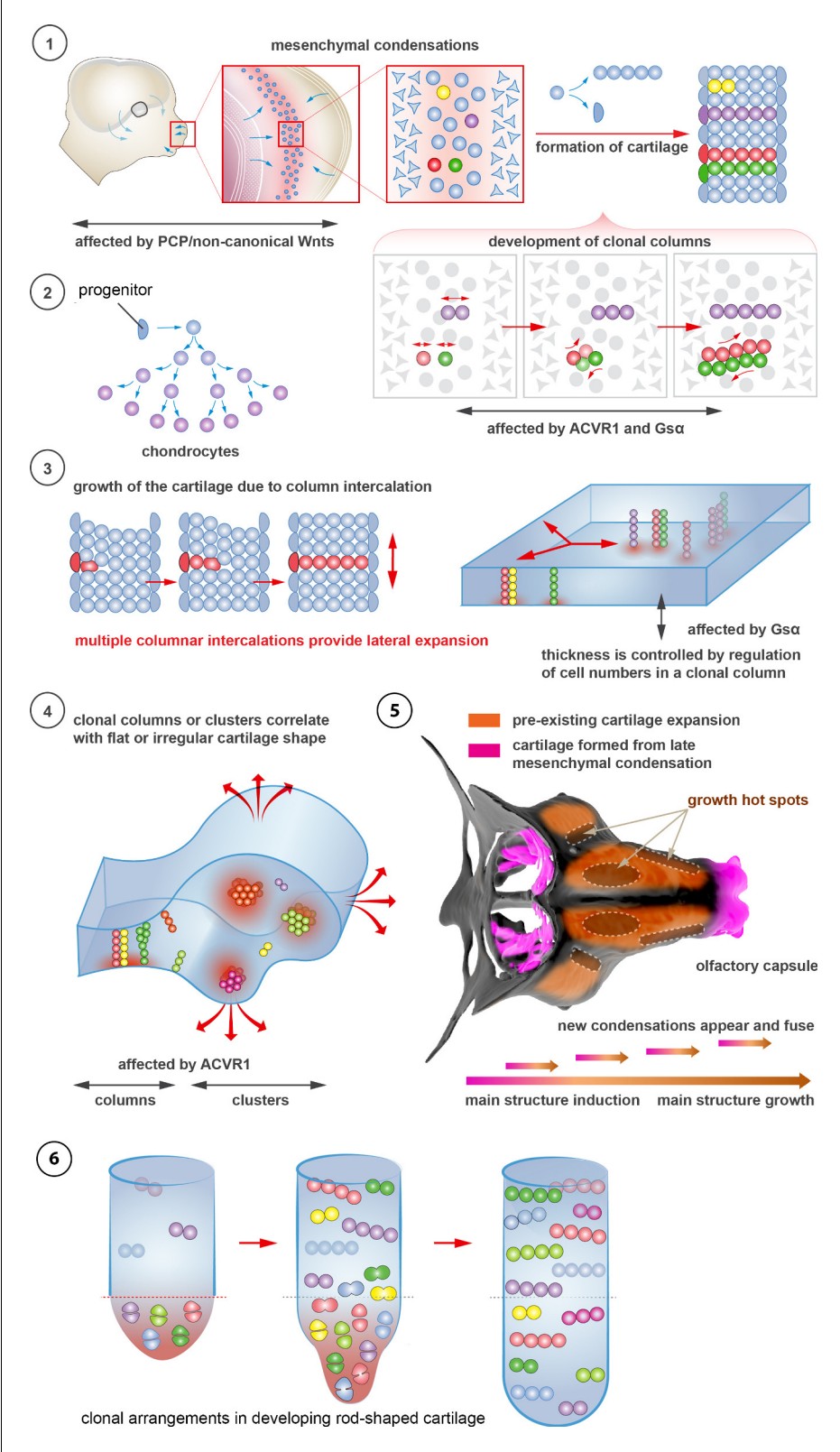

**Figure 10.** Schematic overview of cartilage shaping and scaling processes. (**1**) Oriented cell divisions in the mesenchymal condensations give rise to the transverse columnar clones of chondrocytes. (**2**) Perichondrial cells may potentially give rise to chondrocytes. (**3**) Formation of new clonal columns and their integration into pre-existing cartilage leads to directed lateral expansion of the cartilage. The thickness of the sheet-shaped cartilage depends on the number of cells comprising the column, while the lateral expansion depends on the number of clonal columnar units engaged. (**4**) Geometry of the

*Figure 10 continued on next page*

*Figure 10 continued*

clonal unit corresponds to the overall macro-geometry of the cartilage. Regular clonal units correspond to regular shapes of the cartilage.
(**5**) Chondrogenic mesenchymal condensations are sequentially induced to provide fine details and shape modifications during chondrocranium growth. Upon their maturation, they fuse with the main structure of the chondrocranium. Anisotropic proliferation and specifically positioned proliferative zones further assist the shaping process by imposing physical tensions and curves. (**6**) Rod-shaped cartilage elements also show the regular clonal patterns that result from the transverse orientation of cell divisions and daughter cell allocations that account for the diameter regulation.

are retained until late developmental stages. As we demonstrated, the formation of adjoining mesenchymal condensations occurs in sheet-shaped cranial cartilage. In the developing face, new chondrogenic condensations are responsible for introducing geometrically complicated fine details. Such mechanisms may also operate during amphibian metamorphosis, when most of the postmetamorphic cranial cartilage develops de novo and not from the pre-metamorphic cartilaginous elements (*Kerney et al., 2012*).

Taken together, we discovered important novel principles explaining the growth and shaping of cartilaginous structures. Further studies should focus, amongst other things, on the soluble signals emanating from other embryonic structures which influence the oriented behavior or proliferation of chondrogenic clones.

## Materials and methods

### Mouse strains and animal information

All animal (mouse) work has been approved and permitted by the Ethical Committee on Animal Experiments (Norra Djurförsöksetiska Nämd, ethical permit N226/15 and N5/14) and conducted according to The Swedish Animal Agency´s Provisions and Guidelines for Animal Experimentation recommendations. Genetic tracing mouse strains *Plp1-CreERT2* (RRID:MGI:4837112) and *Sox10-CreERT2* were previously described (*Laranjeira et al., 2011*; *Leone et al., 2003*; *Yu et al., 2013*). *Plp1-creERT2*, *Sox10-creERT2* and *Col2a1-CreERT2* (RRID:IMSR_JAX:006774) (*Nakamura et al., 2006*) (obtained from laboratory of S. Mackem, NIH) strains were coupled to *R26Confetti* (RRID: IMSR_JAX:017492) mice that were received from the laboratory of Professor H. Clevers (*Snippert et al., 2010*). The *Stop^{flowed/floxed}caAlk2-IRES-GFP* strain from the laboratory of Y. Mishina (*Fukuda et al., 2006*) was coupled to *Sox10-CreERT2*. The *Ebf2-CreERT2* (RRID:MGI:4421811) strain was obtained from the laboratory of H. Qian, KI, and was coupled to *R26Tomato*. The *Gsα^{floxed/floxed}* strain was obtained from the laboratory of L. Weinstein (*Sakamoto et al., 2005*). Female mice which were homozygous for the reporter allele [*Gt(ROSA)26Sortm4(ACTB-tdTomato,-EGFP)Luo*/J; Jackson Laboratories] (*Muzumdar et al., 2007*) were coupled to homozygous *Col2a1::creER^T* males [FVB-Tg (Col2a1-cre/ERT)KA3Smac/J; Jackson Laboratories] (*Feil et al., 1997*; *Nakamura et al., 2006*). To induce genetic recombination to adequate efficiency, pregnant females were injected intraperitoneally with tamoxifen (Sigma Aldrich, St.Louis, MO, T5648) dissolved in corn oil (Sigma Aldrich, C8267). Tamoxifen concentration ranged from 1.5 to 5.0 mg per animal in order to obtain a range of recombination efficiencies. Wnt5a, Vangl2 and Ror2 full knock-out embryos were obtained from heterozygous parents (*Gao et al., 2011*; *Yamaguchi et al., 1999*) at the expected Mendelian proportions.

### Immunohistochemistry

For embryo analyses, heterozygous mice of the relevant genotype were mated overnight, and noon of the day of plug detection was considered E0.5. Mice were sacrificed with isoflurane (Baxter, Deerfield, IL, KDG9623) overdose, and embryos were dissected out and collected into ice-cold PBS. Subsequently, the samples were placed into freshly prepared 4% paraformaldehyde (PFA) and depending on the developmental stage they were fixed for 3–6 hr at +4°C on a roller. Embryos were subsequently cryopreserved in 30% sucrose (VWR, Radnor, PA, C27480) overnight at +4°C, embedded in OCT media (HistoLab, Serbia, 45830) and sections cut of between 14 µm to 200 µm on a cryostat (Microm International, Germany), depending on the following application. If needed, sections were stored at −20°C after drying for 1 hr at room temperature, or processed immediately

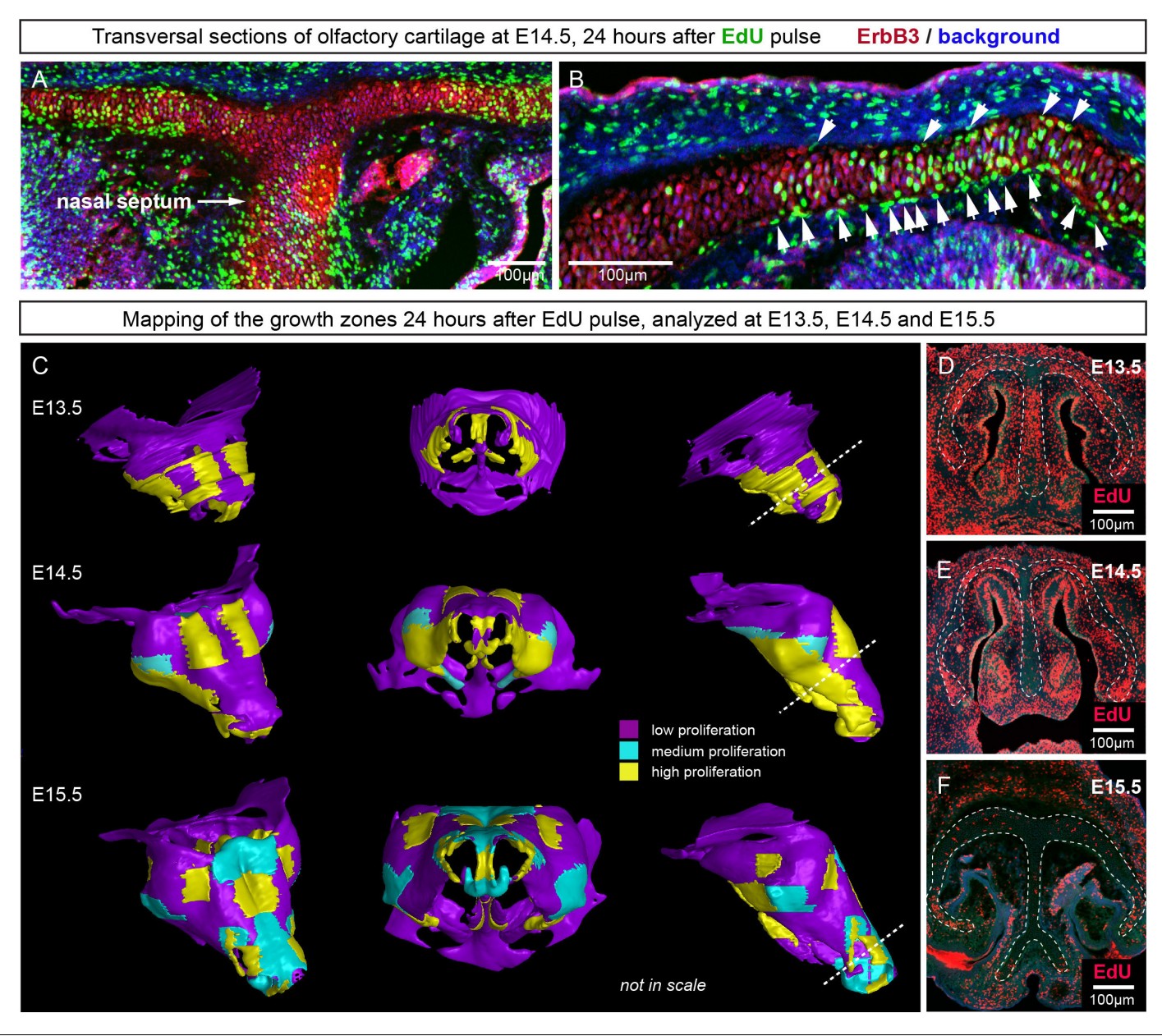

**Figure 11.** Analysis of proliferation identifies specific proliferative regions in nasal capsule. (A–B) Analysis of EdU incorporation 24 hr after the pulse on a transversal section of the facial chondrocranium at E14.5. Notice the distinct proliferative zones in the cartilage that correlate with intense EdU labelling in perichondrial locations shown by arrowheads in (B). (C) Mapping of distinct growth zones onto 3D models of mesenchymal condensations (E13.5) and cartilage (E14.5-E15.5) in the developing face. (D–F) Frontal transversal sections at different developmental stages include proliferative zones within the chondrocranium with EdU incorporation. Scale bars = 100 μm.

after sectioning. Primary antibodies used were: goat anti-GFP (FITC) (Abcam, UK, 1:500, RRID:AB_305635), rabbit anti-Sox9 (Sigma Aldrich, 1:1000, RRID:AB_1080067), rabbit anti-Sox5 (Abcam, 1:500, RRID:AB_10859923), sheep anti-ErbB3 (RnD Systems, Minneapolis, MN, 1:500, RRID:AB_2099728). For detection of the above-mentioned primary antibodies we utilized 405, 488, 555 or 647-conjugated Alexa-fluor secondary antibodies produced in donkey (Invitrogen, Carlsbad, CA, 1:1000, RRID:AB_162543, RRID:AB_141788, RRID:AB_141708, RRID:AB_142672, RRID:AB_2536183, RRID:AB_141844,). Sections were mounted with 87% glycerol mounting media

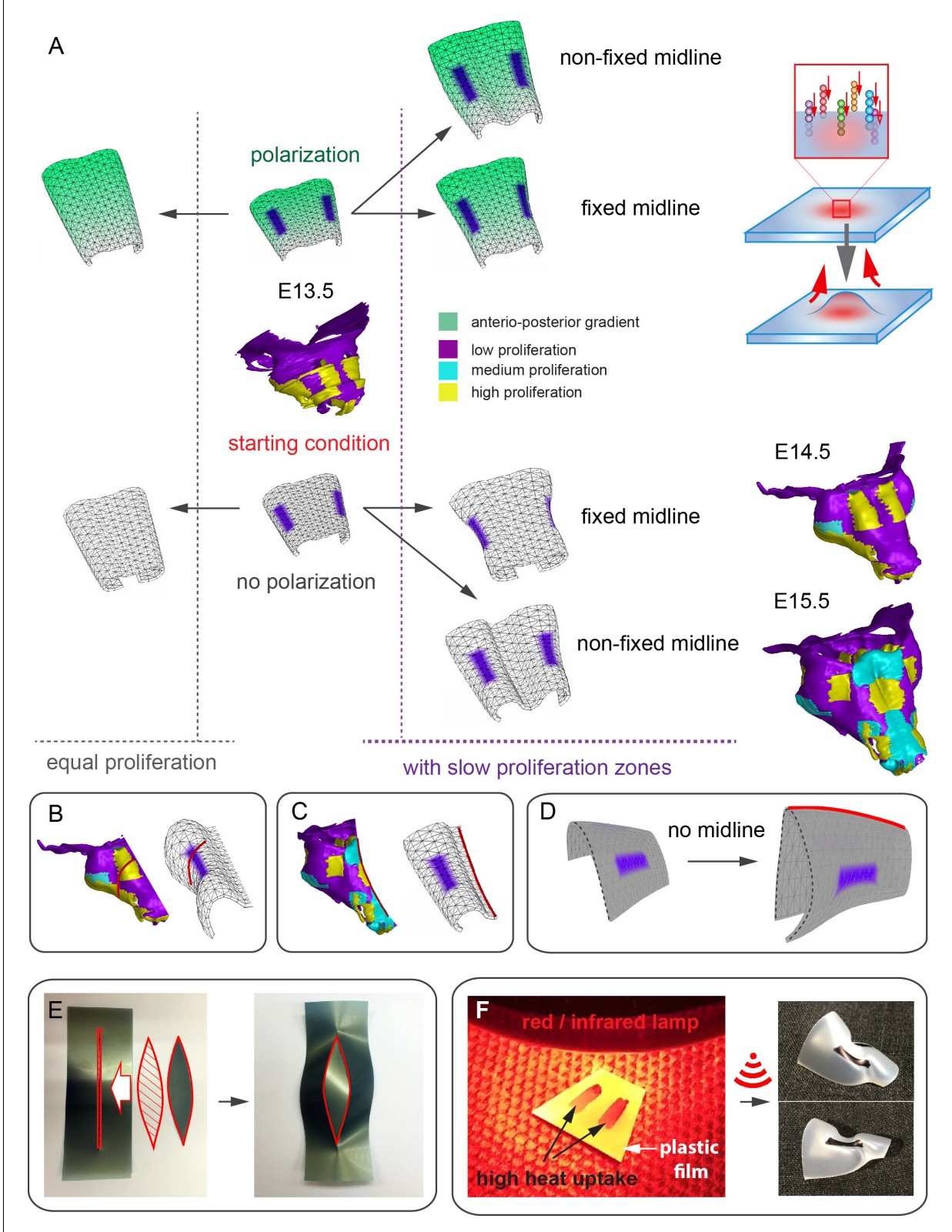

**Figure 12.** Modeling of uneven growth in the shaping of the nasal capsule. (**A**) In silico geometrical transformations of the nasal capsule-like anlage at E13.5 following various scenarios including: anisotropic oriented growth (following polarization introduced by the anterio-posterior gradient shown in green), non-polarized isotropic growth (no anterio-posterior gradient), the presence of fixed midline (simulation of septum and central groove), condition with the unfixed midline (only central groove), conditions with or without slowly growing lateral regions (shown in purple). Note that in
*Figure 12 continued on next page*

*Figure 12 continued*

condition with polarized anterio-posterior growth the anterior elongation of the structure is more prominent and faster as compared to the condition with non-polarized isotropic growth. The lateral bends are induced by slow proliferating lateral regions. In the center and on the right, the real nasal capsules are shown with mapped fast and slow growing regions. (**B**) In the condition with isotropic growth and introduced slowly growing regions, we observe the formation of lateral bends (red line) analogous to the lateral bends in the real nasal capsule at E14.5 (shown on the left). In this condition the midline is fixed, and the ventral groove forms straight. (**C**) In conditions with no fixed midline we observe the formation of the central groove, correct bending of the central groove (red line) and overall flattening of the simulated structure similar to the real object (on the left). (**D**) Simulation with no midline and central groove. Note the inverted bend (red line) and the absence of the correct flattening of the structure. Despite the absence of the midline, the lateral bends are successfully induced by the slow growing regions (purple), analogous to the real nasal capsule. (**E**) Material elastic modelling shows how the third dimension (bending) emerges from changes and tensions in plain 2D structure during imitated anisotropic growth. (E, left panel) Initial modelling conditions: completely flat X-ray film with the cut slot in the middle for fitting the imitated flat growth zone, which is also made from X-ray film. (E, right panel) When the growth zone is inserted into the slit, the whole structure bends to accommodate the tensions. (**F**) Real material (plastic film)-based simulation of isotropic growth was based on uneven shrinking during intense heating. Black painted regions uptake heat more efficiently and shrink faster. The attached edges of the shrinking zone cause bending of the entire structure. Two lateral black stripes were painted on top of the trapezoid as an analog to lateral slowly proliferating zones in nasal capsule. Note the similarity of resulting bends to the lateral bends in real nasal capsule at E14.5.

(Merck, Germany) or in Vectashield Antifade Mounting Medium with DAPI (Vector Laboratories, Burlingame, CA, RRID:AB_2336790).

## EdU incorporation analysis

EdU (Life Technologies, Carlsbad, CA) was injected intraperitoneally into the pregnant females (65 μg per gram of body mass) either 6- or 24 hr before the embryos were harvested. Cells with incorporated EdU were visualized using Click-iT EdU Alexa Fluor 647 Imaging Kit (Life Technologies) according to the manufacturer's instructions.

## Microscopy, volume rendering, image analysis and quantifications

Confocal microscopy was performed using Zeiss LSM710 CLSM, Zeiss LSM780 CLSM and Zeiss LSM880Airyscan CLSM instruments. The settings for the imaging of Confetti fluorescent proteins were previously described (*Snippert et al., 2010*). The imaging of the confocal stack was done with a Zeiss LSM780 CLSM, Plan-Apochromat 3 10x/0.45 M27 Zeiss air objective.

## Histological staining

Slides were stained for mineral deposition using von Kossa calcium staining: 5% silver nitrate solution was added to the sections at a room temperature and exposed to strong light for 30 min. After that the silver nitrate solution was removed, and slides were washed with distilled water for three times during 2 min. 2.5% sodium thiosulphate solution (w/v) was added to the sections and incubated for 5 min. Slides were again rinsed for three times during 2 min in distilled water. The sections were then counterstained using Alcian blue. Alcian blue solution (0.1% alcian blue 8GX (w/v) in 0.1 M HCl) was added to the tissue for 3 min at room temperature and then rinsed for three times during 2 min in distilled water. Slides were then transferred rapidly into incrementally increasing ethanol concentrations (20%, 40%, 80%, 100%) and incubated in 100% ethanol for 2 min. Finally, the slides were incubated in two xylene baths (for 2 min and then for 5 min) before mounting and analysis.

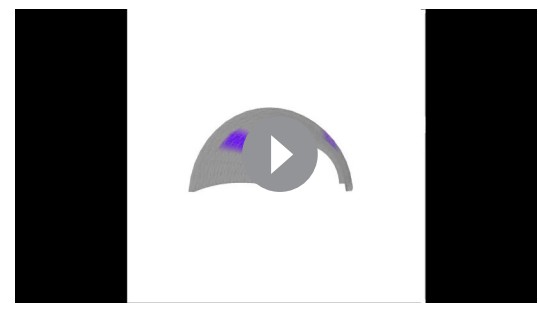

**Video 2.** Simulations of shape transitions of the nasal capsule-like 3D object under different growth conditions. Notice the formation of the lateral bends corresponding to the real nasal capsule shape development from E13.5 to E14.5 occur only in the condition with slow growing purple zones. These bends form independently from isotropic or anisotropic modality of growth.

## Statistics

Statistical data are represented as mean ± s.e.m. Unpaired version of Student's *t*-test was used to calculate the statistics (*P* value). All results were replicated at least in three different animals. Statistical analysis and graphs were produced in GraphPad Prism (La Jolla, CA, RRID:SCR_002798) or Oriana Software (Kovach Computing Services, UK). Spearman coefficient was used for correlation assessment of microgeometries corresponding to different locations in the cartilage.

In *Figure 8* the difference between control (mean = 5.9, sem = ±0.23, n = 4) and mutant (mean = 4.3, sem = ±0.25, n = 3) olfactory cartilage thickness is significant (p=0.0053). The difference between control (mean = 10.6, sem = ±0.83, n = 3) and mutant (mean = 5.7, sem = ±0.61, n = 3) basisphenoid cartilage thickness is significant (p=0.0087).

## Tissue contrasting for μ-CT scanning

Our staining protocol has been modified from the original protocol developed by Brian Metscher laboratory (University of Vienna, Austria). After dissection, the embryos were fixed with 4% aqueous solution of formaldehyde in PBS for 24 hr at +4°C, with slow rotation. Samples were then dehydrated by incubation in incrementally increasing concentrations of ethanol in PBS (30%, 50%, 70%); samples were incubated at +4°C for two days in each concentration to minimize the tissue shrinkage.

We found that the best signal to noise ratio on scans results from contrasting the samples with 0.5–1.0% PTA (Phosphotungstic acid, Sigma Aldrich) in 90% methanol. After sample dehydration, the tissue-contrasting PTA solution was added to the samples and then changed every day with the fresh solution. E12.5 embryos were contrasted with 0.5% PTA for four days while E15.5 embryos were stained in 0.7% PTA for six days. E16.5 and E17.5 embryos were decapitated, and the contrasting procedure was extended to 9–15 days in 1% PTA to ensure the best penetration of the contrasting agent. Subsequently, tissues were rehydrated through a methanol gradient (90%, 80%, 70%, 50% and 30%), to sterile distilled water. After that, rehydrated embryos were embedded in 0.5% agarose gel (A5304, Sigma-Aldrich) and placed in polypropylene conical tubes (0.5, 1.5 or 15 ml depending on the sample size) to minimize the amount of surrounding agarose gel, and to avoid movement artifacts during X-ray computed tomography scanning.

## μ-CT analysis (micro computed tomography analysis)

The μ-CT analysis of the embryos was performed using laboratory system GE phoenix v|tome|x L 240, equipped with a 180 kV/15W maximum power nanofocus X-ray tube and high contrast flat panel detector DXR250 with 2048 × 2048 pixel, 200 × 200 μm pixel size. The exposure time was 900 ms in all 2000 positions. The μ-CT scan was carried out at 60 kV acceleration voltage and with 200 μA X-ray tube current. The voxel size of obtained volumes appeared in the range of 4 μm - 6 μm depending on a size of an embryo. The tomographic reconstructions were performed using GE phoenix datos|x 2.0 3D computed tomography software.

The cartilage of the olfactory system was segmented manually using Avizo - 3D image data processing software (FEI, Hillsboro, OR). The volumetric data of a segmented region were transformed to a polygonal mesh that describes the outer boundary of the region. The polygonal mesh consisting of triangles is a digital geometrical representation of the real object. The polygonal mesh of the olfactory system was imported to VG Studio MAX 2.2 software (Volume Graphics, Germany) for surface smoothening. The analysis of wall thickness at different embryonic stages was performed in order to show the differences or similarities in the thickness of the cartilage structures (*Tesařová et al., 2016*). The results are shown on the polygonal mesh by a colour map. The growth zones in facial chondrocranium at different stages were outlined on top of the 3D polygonal mesh based on the EdU analysis and confocal microscopy results.

## Computer simulations of shape transitions of nasal capsule structure

Models were developed using the growing polarised tissue (GPT) framework and implemented in the MATLAB application GFtbox (*Kennaway et al., 2011*; *Kuchen et al., 2012*) (RRID:SCR_001622). In this method, an initial finite element mesh, also termed the canvas, is deformed during growth. The pattern of deformation depends on growth-modulating factors, whose initial distribution was established during setup. Factors have one value for each vertex and values between vertices are linearly interpolated across each finite element. In the models described here, the initial canvas is

oriented with regard to the external xy-coordinate system such that the canvas base is parallel to the x-axis and the midline is parallel to the y-axis. The initial nasal capsule-line canvas consists of 1800 elements. Elements were not subdivided during the simulations.

Each model has two interconnected networks: the Polarity Regulatory Network (PRN) specifies tissue polarity and hence specified orientations of growth, and the Growth-rate Regulatory Network (KRN) determines how factors influence specified growth rates. In total, growth interactions are specified by three equations, one for the PRN and two for the KRN. These networks determine the specified polarity and growth fields across the canvas. Growth rates are influenced by factors distributed across the canvas. Growth can be promoted in a region by the pro function or inhibited by the inh function as follows:

$$\mathrm{pro(n, x)} = 1 + \mathrm{nx}$$

$$\mathrm{inh(n, x)} = 1/(1 + \mathrm{nx})$$

Due to the connectedness of the canvas, this specified growth differs from the resultant growth by which the system is deformed.

## Fixed midline models

These models question how the structure can transform given that the septum actively anchors the midline and the central groove.

## Setup

The initial set-up phase runs from 0 to 12 time steps and during this phase the canvas deforms from a square sheet into the starting shape for the nasal capsule-like structure. Factor MID is expressed along the proximal-distal midline and used to anchor the midline vertices in the z-plane. Factor CHEEKS is expressed either side of the midline.

## PRN

A proximo-distal polarity field is set up and used to define the orientations of growth. This field is specified as being oriented parallel to the midline throughout growth by the gradient of a polarity factor, POLARISER (POL). POL has a linear gradient across the canvas with the highest level of one at the proximal base and zero at the distal tip.

## KRN

The growth phase occurs after the initial setup phase at time step 13. During this phase there are options for specifying either isotropic growth or anisotropic growth.
During isotropic growth, the growth rate K was set to:
    K = 0.05 · inh(100, iCHEEKS)
During anisotropic growth the specified growth rate parallel to the polarity field, Kpar, was defined as:
    Kpar = 0.05 · inh(100, iCHEEKS)
    while growth perpendicular to the polarity field, Kper, was set to zero.

## Non - Fixed midline models

These models are aiming to simulate what happens to the shape transition when the midline and corresponding central groove are not fixed in space (and can bend or change in any other way) following tensions in the whole simulated structure. We used this approach to question how much the roof of the nasal capsule is anchored by the nasal septum.

## Setup

As with the Fixed-midline model, an initial setup phase runs for 0–12 time-steps in which a square sheet is deformed into an alternative starting shape for the nasal capsule-like structure. In this model the proximo-distal midline was allowed to deform in the z-plane. Factor CHEEKS is expressed either side of the midline and offset slightly distally.

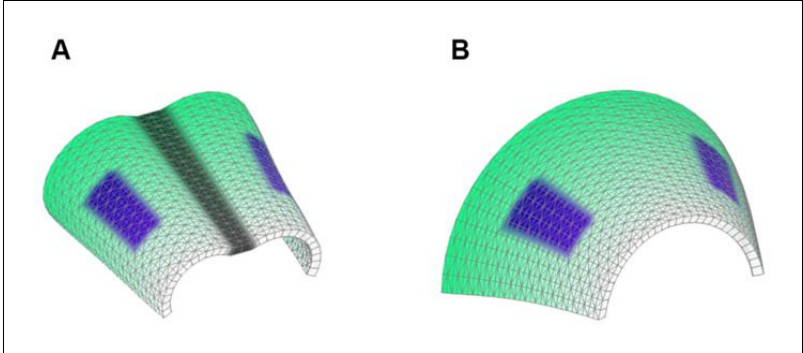

**Scheme 1.** Starting shapes for the Fixed mid-ridge model (A) and Non-Fixed mid-ridge model (B). Green colour indicates the values of POL which is highest at the proximal end. CHEEKS are shown in purple and MID is shown in grey.

## PRN
A proximo-distal polarity field is set up as in the Fixed mid-ridge model.

## KRN
The growth regulatory network is defined as in the Fixed mid-ridge model.

## Mathematical model
For detailed description, please see the Appendix, *Appendix 1—figure 1* and *2* and also (*Kaucka et al., 2016*)

## Acknowledgements
We would like to thank to Olga Kharchenko for illustrations. We are grateful to Prof. Susan Mackem from NIH for the *Col2a1-CreERT2* strain and to Prof. Hans Clevers for the *R26Confetti* strain. M K was supported by EMBO Long-term Fellowship and SSMF (Svenska Sällskapet för Medicinsk Forskning) fellowship. IA and KF were supported by grants from the Swedish Research Council and Karolinska Institutet. IA received support from The Hållsten Research Foundation and Åke Wiberg Foundation. T Z and J K were supported by the Ministry of Education, Youth and Sports of the Czech Republic under the project CEITEC 2020 (LQ1601). J J and A Ha. were supported by the Ministry of Education, Youth and Sports of the Czech Republic (project LQ1605). A He. was supported by the Swedish strategic research programme eSSENCE, the Swedish Research Council (grant #2015–03964) and National Institute of Health under Award Number 1R01EB014877-01. ASC, and PN were supported by the Swedish Research Council (grant # 2016–02835) and Karolinska Institutet. A Deo lumen, ab amicis auxilium. The contents are solely the responsibility of the authors and do not necessarily represent the official views of the National Institute of Health. All authors declare no conflict of interest.

## Additional information

### Funding

| Funder | Grant reference number | Author |
|---|---|---|
| European Molecular Biology Organization | ALTF 216-2013 | Marketa Kaucka |
| Svenska Sällskapet för Medicinsk Forskning | | Marketa Kaucka |
| National Institutes of Health | | Andreas Hellander |

| Svenska Forskningsrådet Formas | Phillip T Newton<br>Andrei S Chagin<br>Kaj Fried<br>Igor Adameyko |
|---|---|
| Karolinska Institutet | Phillip T Newton<br>Andrei S Chagin<br>Kaj Fried<br>Igor Adameyko |
| Bertil Hållstens Forskningsstiftelse | Igor Adameyko |
| Åke Wiberg Stiftelse | Igor Adameyko |

The funders had no role in study design, data collection and interpretation, or the decision to submit the work for publication.

## Author contributions

MK, Conceptualization, Data curation, Formal analysis, Supervision, Validation, Investigation, Visualization, Methodology, Writing—original draft, Project administration, Writing—review and editing; TZ, Software, Formal analysis, Visualization, Methodology, Writing—review and editing; MT, Software, Formal analysis, Validation, Visualization, Methodology, Writing—review and editing; DG, Investigation, Methodology, Writing—review and editing, Responsible for obtaining all Wnt/PCP mutants, sample preparation and help with their analysis; AHe, Resources, Software, Supervision, Validation, Investigation, Methodology, Writing—original draft, Writing—review and editing; JJ, Methodology, JJ has been responsible for tissue contrasting and preparation of the samples for uCT. JJ participated on the review and editing of the manuscript; JK, Supervision, Funding acquisition, Investigation, Writing—original draft, Project administration; JP, PS, Software, Validation, Investigation, Methodology, Writing—review and editing; BS, Validation, Methodology, Project administration, Writing—review and editing, BS has been responsible for preparation of the samples for uCT (tissue contrasting). Participated on data acquiring (confocal microscopy, cryosectioning, staining); PTN, Investigation, Writing—original draft, Writing—review and editing, PTN has been responsible for EdU pulse embryos and for the GSalpha mouse strain. Made very significant contribution to the editing of manuscript (language correction); VD, Investigation, Writing—review and editing, Has been responsible for the production, preparation and analysis of caALK2 embryos; LL, Methodology, Has been responsible for sample preparation, cryosectioning and confocal microscopy; HQ, Conceptualization, Resources, Supervision, Methodology, Writing—review and editing, Preparation of EBF2-GFP/Tomato embryos; A-SJ, Investigation, Methodology, ASJ was responsible for the EBF2-GFP/Tomato strain, injections and sample preparation; YM, Resources, Supervision, Investigation, Writing—review and editing, Responsible for Sox10CreERT2 mouse line; JDC, Data curation, Software, Investigation, Visualization, Methodology, Writing—review and editing; EMT, Resources, Supervision, Funding acquisition, Investigation, Project administration, Writing—review and editing; AE, Software, Investigation, Visualization, Methodology, Writing—review and editing; AD, Software, Supervision, Funding acquisition, Validation, Investigation, Methodology, Writing—review and editing; HB, Resources, Supervision, Funding acquisition, Methodology, Project administration, Writing—review and editing; EC, Resources, Software, Supervision, Funding acquisition, Validation, Investigation, Methodology, Writing—review and editing; MC, Investigation, Methodology, Created and validated the GSa mouse strain; LSW, Resources, Supervision, Project administration, Writing—review and editing; AHa, Resources, Validation, Methodology; EA, Resources, Supervision, Investigation, Writing—review and editing; ASC, Conceptualization, Resources, Formal analysis, Supervision, Investigation, Writing—original draft, Writing—review and editing; KF, Conceptualization, Resources, Supervision, Investigation, Writing—original draft, Writing—review and editing; IA, Conceptualization, Resources, Formal analysis, Supervision, Funding acquisition, Investigation, Writing—original draft, Project administration, Writing—review and editing

## Author ORCIDs

Marketa Kaucka, http://orcid.org/0000-0002-8781-9769
Hjalmar Brismar, http://orcid.org/0000-0003-0578-4003
Enrico Coen, http://orcid.org/0000-0001-8454-8767

Andrei S Chagin, http://orcid.org/0000-0002-2696-5850
Igor Adameyko, http://orcid.org/0000-0001-5471-0356

**Ethics**

Animal experimentation: All animal (mouse) work has been approved and permitted by the Ethical Committee on Animal Experiments (Norra Djurförsöksetiska Nämd, ethical permit N226/15 and N5/14) and conducted according to The Swedish Animal Agency´s Provisions and Guidelines for Animal Experimentation recommendations.

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

## Appendix 1

# Individual based model (IBM) for cartilage dynamics

In order to model and illustrate the growth of the cartilage on the cellular level, we developed an individual-based model incorporating cell proliferation (including displacement of surrounding cells via pushing) (*Hellander, 2015*). The model is stochastic because we want to be able to capture effects of e.g. synchronicity in cell division and the degree of determinism needed to achieve and ordered columnar growth of the structure. We do this by letting the cell division times, direction of allocation of daughter cells after cell division, etc., be random variables. This document describes the details of the model and its implementation.

There are several popular modeling frameworks for simulating interacting cells. In the cellular potts model (*Graner and Glazier, 1992*) a single biological cell can be composed of multiple lattice sites making it possible to use a more detailed description of cell shape and include more detailed description of more mechanical properties. In off-lattice center based models cells are often modeled as spheres with pair-wise interactions and a force-based description to evolve the system dynamics, for an overview see (*Van Liedekerke et al., 2015*). Vertex models can offer even more realistic models of cell mechanics (*Fletcher et al., 2013*) but become expensive and complicated in three space dimensions.

Rather than these more comprehensive mechanical models, we use a simplistic rule-based, on-lattice stochastic cellular automaton (CA). In the language of a recent review (*Van Liedekerke et al., 2015*) our model falls into the category of a Type B CA. These types of models are widely used in e.g. cancer tumor modeling and for simulation of monolayers and spheroids (*Radszuweit et al., 2009*). The simulation code is written in Python, relies on the PyURDME package for spatial stochastic simulations (www.pyurdme.org) and it is freely available for download from www.github.com/ahellander/multicell (*Hellander, 2015*) under the GPLv3 license. A copy is archived at https://github.com/elifesciences-publications/multicell.

The basic entities in our simulation are Agents and Events. An Agent is a model (implemented as a Python class) of an individual (cell). Events simulate discrete state changes involving one or several agents. They occur at a certain time (assuming no other event involving the same agents occurs first). They rely on rules that specify how and under what conditions the event is to be executed. A simulation is initialized by creating the initial population of agents and events, and then creating a priority queue (in our case implemented using a heap data structure using the Python module' heapq'). In each iteration of the algorithm, the event with the shortest time is popped from the queue and executed (assuming that all of its rules and conditions can be satisfied), the system time updated, new events derived from any newly created agents are created and inserted into the queue, and all existing events affected by changes in the agents or the system state are updated.

Each agent occupies one voxel of a tessellation of 2D or 3D space, and each lattice site can only accommodate one agent. Following the recommendation in (*Radszuweit et al., 2009*) a Delaunay triangulation is used. The mesh resolution is chosen such that the average voxel size is close to the desired cell size (~7 μm radius) taken from the experiments. Being a lattice model, the shape and volume of the cells are a lattice property and are given by the dual grid (Voronoi cells in the case of a Delaunay triangulation). This is illustrated in *Appendix 1—figure 1A*. The interpretation is that individual agents occupy the dual elements (dashed lines) of an unstructured triangular (2D) or tetrahedral primal mesh (3D) (solid lines).

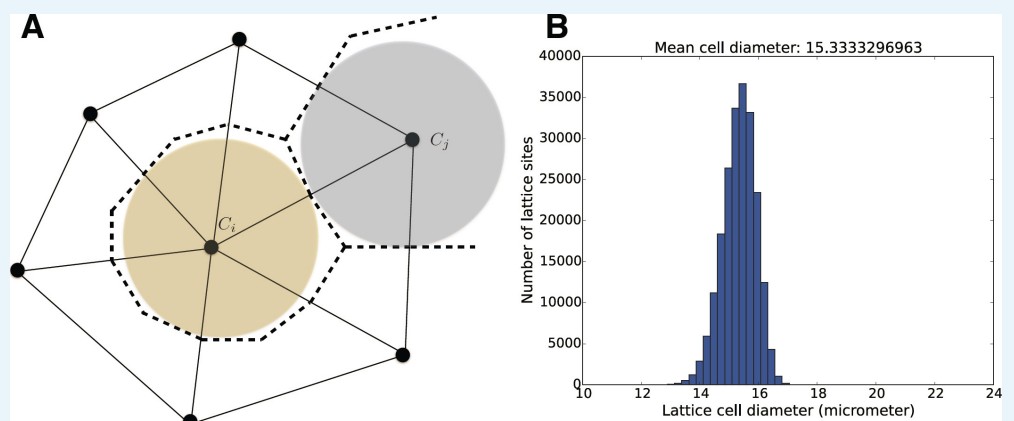

**Appendix 1—figure 1.** (A) Individual cells are modeled by a number of properties such as their color and distributions for cell division times. The positions of cells in space are tracked on an underlying unstructured lattice, or grid. The edges in the primary mesh (solid lines) connect vertices (black dots). A biological cell is modeled by the volume made up of the dual elements (dashed lines), connecting triangle (2D) or tetrahedral (3D) centers and edge or face centers. For visualization purposes, in 3D space, we plot cell individuals as colored spheres with radius equal to the sphere with equal volume as the dual element. (B) Size distribution for the mesh elements for the geometry and mesh used in the simulations in the *Figure 5*.

## The individual agents – colored coded cells

Each individual cell is modeled as an individual agent with the following properties:

- Color (a label used to track the lineage).
- Mean cell division time, $\mu_p$.
- Variance in cell division time, $\sigma_p^2$.

When visualized in 2D, we draw cells as polygons (the actual dual cells) and in 3D for practical reasons we visualize them as spheres centered on the vertices of the primal mesh, with radius chosen such that the volume corresponds to the volume of the corresponding dual element. On the unstructured mesh, there will be a size distribution for the mesh elements, i.e. there will be a small variation in the size associated with each lattice site, see *Appendix 1— figure 1B*.

## Cell proliferation

### Cell division time

The time until a cell, or individual, divides, is assumed to be a random variable. Although a multi-stage model of cell division can give rise to an an Erlang distribution (*Radszuweit et al., 2009*) which is found to match experimental data for another system, we are not calibrating our model to experimental data on cell division time distributions. The dividing cell (referred to as the mother cell) create a daughter cell after a normally distributed waiting time $\tau_D$.

$$\tau_D \sim N\left(\mu_d, \sigma_d^2\right) \tag{1}$$

where the mean division time $m$ and the variance $\sigma^2$ are parameters of the model to be supplied as input to the simulation. As a measure of the degree of variability in cell division times, we use the standard deviation over the mean,

$$f = \frac{\sigma_d}{\mu_d} \tag{2}$$

The smaller value of $f$, the more deterministic and synchronized the cell cycles of individual cells are. The way our model is set up, there are no events that leads to the recalculation of a cell's division time. A cell gets assigned a time to division at creation and each cell then divide according to its internal clock irrespective of if it gets pushed etc.

After division, the daughter cell needs to be deposited on the grid. The division direction, or receiving voxel, is sampled according to a discrete distribution. In the simplest case, all directions of division are equally probable and the direction distribution is uniform. In the general case, weights are assigned according to an external, deterministic gradient.

## Cell division direction

The division direction is also a random variable and the number of possible directions are given by the connections to the neighbors on the grid. Each individual has a property that sets its polarization, represented by a normalized vector $p$ pointing in the preferred direction of division. With no polarization, each possible division direction is equally probable. With polarization, the probability to divide in a certain direction is biased by the gradient. The weights for sampling the division direction are taken to be

$$w_{ij} = \left( \frac{d_{ij}}{max_j(d_{ij})} \right)^b, \tag{3}$$

$$d_{ij} = \frac{g(x_j) - g(x_i)}{h_{ij}} \tag{4}$$

where $x_i$ is the position of the vertex in the grid for which the agent resides, and $h_{ij}$ is the length of the edge connecting grid points $x_i$ and $x_j$ and $g(x)$ is a given concentration profile. The parameter $b \geq 0$ dictates how perfectly the cells become polarized by the concentration profile $g(x)$. A value $b = 0$ leads to equal probabilities for all directions, and very large value of $b$ means that the division direction will always be in the direction of the maximal value of the gradient (the division direction becomes deterministic in the direction of the maximal gradient in the limit $b \to \infty$. Values in between the extremes describes an increasing precision in polarization axis alignment with the gradient field.

## Cell pushing

If the receiving lattice for the daughter cell site is empty (i.e. occupied by matrix), it is simply deposited there. When the daughter cell cannot be placed on a free lattice site, there is an attempt to reorganize the structure by pushing neighboring cells to make room for it. The procedure is illustrated in *Appendix 1—figure 2*. The probability for the displaced cell to move to a given neighboring grid point depends on the direction of pushing. Let $e_{md}$ be the vector along the edge connecting the mother cell $C_m$ and the daughter cell $C_d$, pointing towards the daughter cell. Let $e_{dk}$ be the unit vector along the edge connecting the daughter cell $C_d$ and one of its neighbors, $C_k$. The weight for moving the displaced cell to the neighbor with index $n$ is given by

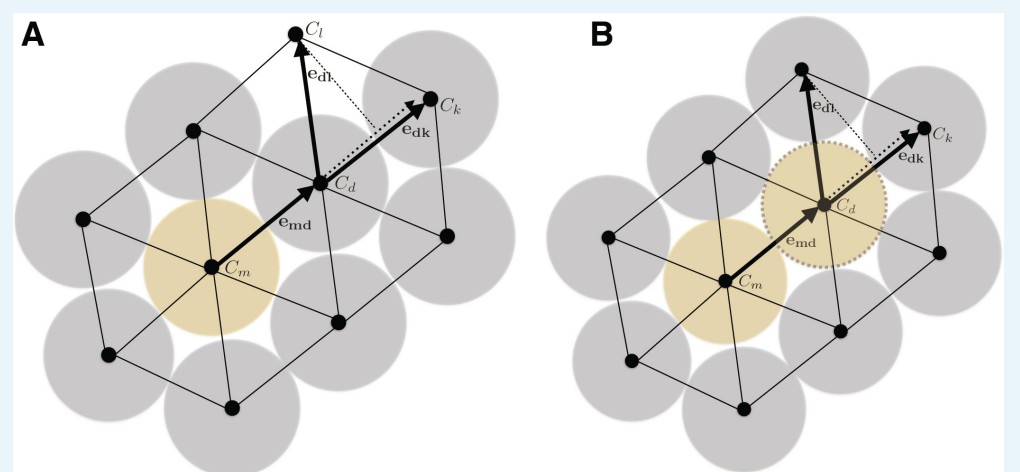

**Appendix 1—figure 2.** When a cell divides (yellow cell), the daughter cell will push surrounding cells (blue) to make room for the progeny (**A**). The direction of pushing (and what cell to push) is determined by a combination of the directivity of the original division or pushing direction and of a penalty for pushing an occupied site. The penalty is governed by a parameter c that dictates how much to favor pushing into a free lattice site. In the figure, pushing the blue cell at location $C_d$ to $C_l$ will be favored over $C_k$ based on the penalty if $C_k > 0$, but the site $C_k$ if favored based on the directionality of the push. If $C_k$ is selected by the probabilistic algorithm, it will in turn complete a pushing event, leading to a pushing chain that continues until a cell is pushed into an empty site. After such a pushing chain has been completed, the site sampled for the daughter cell will be free, and the newly created daughter cell can be inserted (**B**).

$$w_k = max(s_k(1 - cI_n), 0) \tag{5}$$

where

$$s_k = e_{md} \cdot e_{dk} \tag{6}$$

That is, to account for the directionality implied by the originally sampled division direction $e_{md}$, the probability of the pushed cell to move to an adjacent lattice site $c_k$ is proportional to the scalar projection of the vector connecting the pushed cell and the vertex at index k onto the direction vector of the push. $c$ is a penalty parameter in the interval *[0, 1]* that dictates the degree of resistance in pushing the cell into an already occupied site. If $c = 1$, it is not possible to push a cell into an occupied site, and if $c = 0$, there is no resistance what so ever and only the direction of the push affects the displacement direction. Any value in between is a tradeoff between the two extremes. In *Appendix 1—figure 2*, for example, there is a high probability to push $C_j$ to $C_k$, due to $e_{dk}$ being almost parallel to $e_{md}$ (high $s_k$), but depending on the value of $c$, the site $C_k$ may be sampled instead since that site is empty, even if the directionality contributions is smaller ($s_l < s_k$). Once a neighbor has been displaced, the pushed cell moves into that lattice site and the daughter cell gets deposited on the now free lattice site of the displaced cell. The displaced cell, may in turn then go on to displace additional cells and this procedure is repeated until a cell gets pushed into an unoccupied site.

## Measure of order in the cartilage model

We are interested in assessing what factors are the main determinants to the degree of ordered columnar growth in the cartilage sheet patches. To that end, we postulate that a perfectly ordered structure consists of clonal columns growing straight and directed along the axis perpendicular to the initial condition starting plane (*Figure 5G*). We then use the following metric to quantify the degree of order in the structure

$$S = \frac{1}{C}\sum_{j=1}^{C}\frac{1}{N_j}\sum_{i=1}^{N_j}s_{ij} \tag{7}$$

with $s_{ij}$

$$s_{ij} = |v_i \bullet y| \tag{8}$$

where $y$ is a unit vector perpendicular to the initial condition plane, and $v_i$ are normalized vectors joining two consecutive points (sorted by y-coordinate) in clones with the same color. $N_j$ is the number of cells of a given color minus one (the number of vectors), and $C$ is the number of unique clones tracked. This is illustrated graphically in *Figure 5A*. With this metric, a score of $S = 1$ would mean that all columns are perfectly aligned to the main growth axis and a score of $S = 0$ would mean that they are all perpendicular to it. We use this metric to score realizations of the process either in the absence of a gradient, or when the gradient is uniform in planes parallel to the center plane, so that in the case of a perfect polarization, cells should all deposit their daughter cells perpendicular to the center plane.

