## [Decision Letter]

[Editors’ note: a previous version of this study was rejected after peer review, but the authors submitted for reconsideration. The first decision letter after peer review is shown below.]

Thank you for submitting your work entitled "New growth mechanism operates in facial cartilage scaling and shaping" for consideration by *eLife*. Your article has been reviewed by three peer reviewers, and the evaluation has been overseen by a Senior Editor. The reviewers have opted to remain anonymous.

Although all of the reviewers find the question very interesting and many aspects of the manuscript well done, they also raise numerous substantial issues that preclude publication in *eLife*. After consultation between the reviewers, it was deemed that the required revisions were so substantial that it would take a great deal of time and effort to produce a modified version that would meet their concerns. Based on these discussions and the individual reviews detailed below, we regret to inform you that your work will not be considered further for publication in *eLife*.

*Reviewer #1:*

The authors of the manuscript "New growth mechanism operates in facial cartilage scaling and shaping" analyzed growth of facial, specifically nasal/snout cartilages and proposed a mechanism by which it expands. More specifically, they suggest that facial cartilage growth that is based on introducing oriented clonal cell units into existing cartilage from the perichondrial cell layer. While the paper is interesting and makes a number of important observations, in its current form it is not very clearly written with some parts redundant, it makes a number of statements which need further clarification or support in the form of additional experiments. The paper draws conclusions which seem go beyond the most obvious interpretation of their data and to be more convincing of the main points the paper is trying to make. The significance of cranial synchondroses in facial cartilage growth is overlooked and while contribution from clonal expansion to such cartilages has not been reported before, such mode of growth is not new and its significance may have been exaggerated.

*Reviewer #2:*

Kaucka et al. paper aimed to address two important questions of craniofacial skeletal growth: 1), how does chondrocyte cellular behavior contribute to cartilage shaping and scaling; 2), what are the progenitors of chondrocytes? The authors started with characterizing tissue growth using micro-CT and found that tissue expansion was anisotropic mainly along its surface area (lateral) but not the short axis (thickness/depth). Interestingly, in confetti mice, chondrocytes clones were arranged in columns parallel to the short axis of the tissue (depth or width?). Combing this observation with mathematical modeling, the authors predicted that the polarized growth of columns controls tissue thickness. To test this hypothesis, they perturbed the activity of multiple signaling pathways and found an association between clone morphologies and tissue dimensions in mutant conditions. Next, to determine the lineage relationship between chondrocytes and perichondrium, the authors characterized clone distribution in confetti mice. In parallel, they performed H2B-GFP pulse-chase to analyze the distribution of daughter cells after division. In both cases, the labeled chondrocytes and perichondrial cells were clonally related, which made the authors believe that chondrocytes are derived from perichondrial cells. Finally, in silico modeling suggests that the diversity of cartilage morphology is driven by the uneven distribution of proliferative chondrocytes. Their work provides list of insights including:

1) Directional growth of cellular columns controls the thickness of craniofacial cartilage

2) Chondrocytes are derived from perichondrial cells

3) Uneven proliferative zone drives cartilage shape turning

This paper combined powerful mouse genetics with computational modeling and tomographic reconstruction to address the fundamental questions in morphogenesis. However, the analysis is not rigorous and the data is overinterpreted.

1) Clonal analysis is not clear at spatial level. The 3D tomographic reconstruction revealed that the surface size of cartilage increased significantly while thickness (depth) remained constant. However, the clonal analysis was performed at 2D level. The authors need to show if the direction of column growth is mainly along the width or the depth of the tissue. This is critical to determine how cellular organization contributes to 3D morphogenesis of the tissue. Additionally, in Figure 3, the authors developed a quantitative method to characterize clone morphology in wild-type and caALK2 mice, this approach should be applied to other studies (Figure 1, Figure 1, Figure 2, Figure 5).

2) Clonal analysis is not clear at temporal level. Tomographic reconstruction showed that tissue thickness increased from E14.5 to E16.5 but remained constant from E16.5 to E17.5. If tissue thickness is driven by column expansion, the size of columns should show similar growth kinetics. This hypothesis needs to be tested.

3) The tomographic analysis is quantitative in wild-type tissues (Figure 1) but descriptive in BMP and PCP mutant mice. To draw the conclusion that cartilage display abnormal morphologies, the same quantitative analysis needs to be performed on mutants.

4) The authors identified a correlation but not a causal linkage between cellular columns and tissue dimensions. caALK2 mice were driven by *Sox10-Cre*, which interfered the normal activity of both cartilage and adjacent perichondrium. Thus, it is possible that cartilage shaping is not controlled by chondrocyte column formation; instead, both of these two factors are affected by neighboring tissues. Supporting this possibility, musculature contraction is critical to chondrocyte organization and skeletal morphogenesis (Shwartz et al., 2012, Developmental Biology). Similarly, all the PCP mutant mice were full knock-out, which makes it difficult to interpret the phenotype in cartilage. Gsa mutant mice are driven by Col2-Cre that is more chondrocyte specific, but the adjacent perichondrium were still labeled (Figure 5). In this case, the possible non-autonomous function of perichondrium could not be ruled out.

5) Some other questions about Gsa mutant mice: The authors found that these more matured cells divided more slowly. But the clonal size appears similar (compare 5B and C, compare 5D and E). Were the mutant proliferative cells differentiated into hypertrophic cells? In the growth plate of long bones, these cells are larger than proliferative cells. Could these matured cells affect cartilage morphology in Gsa mutant? Since the lateral expansion of the tissue was not measured, how could the authors know cell division only affects tissue thickness?

6) The progenitor role of perichondrium is uncertain. Based on the spatial distribution of clonally related cells, an alternative hypothesis is that the perichondrial cells are derived from chondrocytes. Because this lineage relationship is an important part of the paper, it is desirable to perform lineage-analysis using perichondrium specific Cre or time-lapse imaging to follow cell behaviors in the short-term. Time-lapse imaging on mouse craniofacial skeletons is technically possible (Romerim et al., 2014, Development).

*Reviewer #3:*

This paper poses an interesting question, how do cartilaginous structures such as the nasal capsule grow without evidence of set growth plates, as in long bones etc.?

Figure 1 clearly highlights this problem by showing that the pattern of the nasal cartilages are set around E14.5 but that substantial growth (scaling) occurs during later embryonic stages.

The question in tackled by following clones of cells in the cartilage revealing the formation of orientated doublets and lines of cells, which are very striking observation. This is a really interesting result that is then backed up by following patterns of clones in cartilages of different shapes and in different mutants.

Some parts of the paper are not as clear as they could be, with some experiments not explained and a few things presumed.

It is not clear why the authors have used PLP, Sox10 and Col2 cre mice. The Col2 appears to be enough to show the results. Are their different advantages of the different lines, or is there a benefit in targeting slightly different populations?

The authors call their cartilages flat. I am not sure this is a good description of cartilages that are in complex 3D sheets. Perhaps their definition can be explained further. Do they mean a smooth surface?

I cannot comment on the model presented from a mathematical point of view but the graphics are generally very good.

In the ACVR1 experiments clumps are shown at the periphery of the cartilages. This is a very nice clear result. In these mice are the cartilages shorter, as would be predicted if the cells were failing to intercalate successfully? In Image N there are quite a few GFP cells that might be expected to have altered the intercalation in this part of cartilage.

The PCP mutants are interesting to illustrate the difference between initial patterning defects and problems with later scaling. Likewise the Gs-α experiments separate the role of integration from proliferation for controlling cartilage thickness.

In the early section of the text the authors suggest that the cartilage cells move in from the perichondrial layer, seeding the vertical columns. The origin of the new cartilage cells is an important point and is tested using dilution of H2B-GFP. I find this experiment the weakest in the paper as it does not clearly prove a perichondrium origin, but simply reiterates that the perichondrium and cartilage cells are clearly clonal. The graph shows that the perichondrium appears to proliferate more than the cartilage cells but this is not showing point of origin. Time lapse would be a clear way of proving the origin of the cells.

The last part of the paper deals with shape changes during later development. A key part is an analysis of bending but do the cartilages actually bend? From the Introduction I thought the nasal cartilage was simply scaling from an original pre-cartilage template with minor adjustments. The cartilage is not a sheet but already curved. If bending does occur this should be shown very clearly using real images, rather than the schematic shown. The sections of the nasal region in the supplementary data do not give much impression of bending, perhaps at the bottom of the capsule?

Confirmation of bending is necessary to confirm that modelling on flat plastic is relevant. The use of actual materials, however, is a very nice way to visualise morphogenetic movements, following from the work of Enrico Coen on plant morphogenesis.

[Editors’ note: what now follows is the decision letter after the authors submitted for further consideration.]

Thank you for resubmitting your work entitled "Oriented clonal cell dynamics enables accurate growth and shaping of cartilage" for further consideration at *eLife*. Your revised article has been favorably evaluated by Marianne Bronner as the Senior Editor and Reviewing Editor, and two reviewers.

The manuscript is much improved but there are some remaining issues that need to be addressed by alterations in the text prior to acceptance, as outlined below:

1) With respect to whether chondrocytes function autonomously to control cartilage morphology, how could the authors exclude the possible roles of this layer? They argue that for an autonomous effect based on observing that cells with recombined GPF express Sox9 (Figure 6). However, there are two layers of perichondrium: and inner layer that expresses Sox9 and an out layer that expresses collagen type I. Could this outer layer cause a non-autonomous effect in cartilage shaping? The outer layer perichondrium also contains recombined cells in both confetti mouse (Figure 6) and caALK2 mouse (Figure 6), both of which are driven by Sox10 promoter. Therefore, the possibility of non-autonomous effects should be discussed.

2) Kudos to the authors for trying very hard to do live imaging to test the lineage relationship between chondrocytes and perichondrium. However, the "n" (Figure 9) is very low. Given that the total number of cell division drops in organ culture, such that they cannot perform statistical analysis, they should leave out this data.

3) In Figure 9, the authors use salamander regenerating limb as a tissue model for live imaging. However, cell behaviors could be very different during normal development versus regeneration. In addition, it previously has been shown that during cartilage regeneration, perichondrium cells can regenerate chondrocytes (Srour MK et al., Natural large-scale regeneration of rib cartilage in a mouse model, 2015). Therefore, these data may not be relevant to the story and the authors may want to leave it for a different paper.

4) The paper would profit from some rewriting. Parts of the paper are not well organized; the Summary and Introduction are too long and some of the contents seem to be misplaced; there are many grammatical mistakes.

---

## [Author Response]

[Editors’ note: the author responses to the first round of peer review follow.]

*Reviewer #1:*

*The authors of the manuscript "New growth mechanism operates in facial cartilage scaling and shaping" analyzed growth of facial, specifically nasal/snout cartilages and proposed a mechanism by which it expands. More specifically, they suggest that facial cartilage growth that is based on introducing oriented clonal cell units into existing cartilage from the perichondrial cell layer. While the paper is interesting and makes a number of important observations, in its current form it is not very clearly written with some parts redundant, it makes a number of statements which need further clarification or support in the form of additional experiments. The paper draws conclusions which seem go beyond the most obvious interpretation of their data and to be more convincing of the main points the paper is trying to make. The significance of cranial synchondroses in facial cartilage growth is overlooked and while contribution from clonal expansion to such cartilages has not been reported before, such mode of growth is not new and its significance may have been exaggerated.*

We are very grateful to the reviewer 1 for pointing out these issues and various questions, specifically about synchondroses and mechanisms of cartilage growth in general. We heavily worked on them many months during this revision. Mostly, we were very inspired by suggestion to analyze the role of synchondroses in cranial base and nasal capsule expansion. In this revision, we provided additional data addressing both synchondroses and apical growth zone. Please see new Figure 1, Figure 2 and three associated supplementary figures. In these figures, we show the visualizations of synchondroses development and, despite it has an immense role in cranial base expansion, we proved that synchondroses, as well as direct endochondrial ossification processes, do not happen in the nasal capsule and many other parts of the chondrocranium during the period of embryonic development that we study here. And still, these cartilaginous parts are growing quite isotropically, while synchondroses provides rather unidirectional (anterio-posterior) expansion of the base of the skull. We showed, explained and discussed (with literature citations suggested by the reviewer 1) the importance of synchondroses. Our findings are clearly demonstrated in Figure 1. In addition to CT and histology-based approach, we also analyzed clonal cell dynamics in classical synchondroses (right part of Figure 1, panels D-I) and we never see such cell dynamics during the development of the nasal capsule in an embryo.

We also followed the advice of the reviewer and birth-dated the cartilage with inducible genetic tracing approach and showed that cartilage born from the apical growth zone and cartilage formed from newly induced mesenchymal condensations elsewhere do not explain fully the isotropic growth of the nasal capsule (please see new Figure 2 and associated supplementary figures). These results are completely new and have not been reported before. Our new data are presented in a new Figure 2 and 2 new supplementary figures. They show that mesenchymal chondrogenic condensations are continuously produced at the tip of the nasal capsule and around the nasal conchae. Later, this freshly produced cartilage fuses with the main nasal capsule structure to create the fine geometrical features. At the same time, the basic geometry of nasal capsule keeps growing. Thus, the third mechanism is taking place in addition to synchondroses, new chondrogenic condensations and apical growth zone.

When we were writing the first version of our manuscript, we did not aim to diminish the other ways of cartilage growth including synchondroses. We are happy that reviewer 1 helped us to obtain new results and inspired to formulate and discuss them more correctly in this light.

The reviewer 1 writes “such mode of growth is not new”. We hope that this statement results from some pedagogical/writing style-related issues from our manuscript. We focus on the geometry of clonal arrangements, anisotropic proliferation, oriented cell behavior, mechanical tensions and other mechanisms of growth and shaping of the facial cartilage in this manuscript. We are not aware of a publication describing these new clonal cell dynamics during embryonic chondrocranium growth elsewhere. We hope that after serious rewriting of the manuscript and doubling the amount of data during this revision, these problems have dissipated, and the reviewer will find our response well-grounded in the new results and comprehensive.

*Reviewer #2:*

*[…] This paper combined powerful mouse genetics with computational modeling and tomographic reconstruction to address the fundamental questions in morphogenesis. However, the analysis is not rigorous and the data is overinterpreted.*

*1) Clonal analysis is not clear at spatial level. The 3D tomographic reconstruction revealed that the surface size of cartilage increased significantly while thickness (depth) remained constant. However, the clonal analysis was performed at 2D level. The authors need to show if the direction of column growth is mainly along the width or the depth of the tissue. This is critical to determine how cellular organization contributes to 3D morphogenesis of the tissue.*

We completely agree with this comment. In the revised version of the manuscript we provide these new data on the 3D clonal column shape in Figure 4—figure supplement 2, new panel A, and also about clonal column growth in Figure 3—figure supplement 2, new panel E.

*Additionally, in Figure 3, the authors developed a quantitative method to characterize clone morphology in wild-type and caALK2 mice, this approach should be applied to other studies (Figure 1, Figure 1, Figure 2, Figure 5).*

Indeed, we quantified the clone morphology in facial cartilages of the wild type mice and caAlk2 mutants. However, in the old Figure 1 we describe micro-CT data, and thus we do not understand what reviewer 2 means here. Figure 2 shows the situation in the wild type mice, so the proposed quantification will be obviously redundant since it was already done and presented in old Figure 3 (new Figure 6). Old Figure 5 (new Figure 8) shows Gsa phenotype also with genetic tracing (oriented columns and doublets are visible), and we do provide the requested additional analysis of clonal shapes in new Figure 8. Importantly, facial shape does not change substantially in these mutants and, as we prove with newly supplied analysis in Figure 8, does not involve changes in clonal orientations. We apologize that this was not obvious in the first version of the manuscript.

*2) Clonal analysis is not clear at temporal level. Tomographic reconstruction showed that tissue thickness increased from E14.5 to E16.5 but remained constant from E16.5 to E17.5. If tissue thickness is driven by column expansion, the size of columns should show similar growth kinetics. This hypothesis needs to be tested.*

We thank our reviewer for this advice and we addressed this comment. Indeed, eventually columns increase cell numbers between developmental stages. This new result can be found in Figure 3—figure supplement 2, new panel E.

*3) The tomographic analysis is quantitative in wild-type tissues (Figure 1) but descriptive in BMP and PCP mutant mice. To draw the conclusion that cartilage display abnormal morphologies, the same quantitative analysis needs to be performed on mutants.*

We do show the quantitative analysis for the Alk (BMP receptor) mutants (new Figure 6) when it comes to clonal shapes. We cannot do the overall thickness analysis of the entire nasal capsule because the genetic penetrance (efficiency of Alk2 activation controlled by genetic tracing) is generally very low – too few clones appear to be affected by genetic recombination (and those we can efficiently observe). However, we show quantitative thickness analysis of the most affected regions (new Figure 6, please see the insets with quantified thickness heatmap). Indeed, the microgeometries are affected as demonstrated in new Figure 6, while the majority of normal unaffected cartilage provides for the general recovery of the nasal capsule macro-shape. The suggested analysis of nasal capsule from caAlk2 mutants (similar to the one reported in the old Figure 1 (new Figure 3)) will not make any sense – most of the measured positions will reflect only the wild type cartilage.

When it comes to the analysis of PCP mutants: in the new revised Figure 7 we introduced the quantitative analysis of thickness following this reviewer comment.

Importantly, microgeometries of a sheet-like cartilage are fine in all those mutants and do not show neither geometrical nor clonal orientation-related differences from the control. This is the only aspect that is truly important in a context of this study. For this we demonstrated quantitative EdU-based analysis of clonal orientations. Other panels simply outline the general picture and stand for the pedagogical purpose, while the entire figure shows a clear negative result at the level of clonal structure/orientations etc.

*4) The authors identified a correlation but not a causal linkage between cellular columns and tissue dimensions. caALK2 mice were driven by Sox10-Cre, which interfered the normal activity of both cartilage and adjacent perichondrium. Thus, it is possible that cartilage shaping is not controlled by chondrocyte column formation; instead, both of these two factors are affected by neighboring tissues. Supporting this possibility, musculature contraction is critical to chondrocyte organization and skeletal morphogenesis (Shwartz et al., 2012, Developmental Biology).*

We disagree with this reasoning because all recombined cells in case of caALK2 experiment turned into the Sox9^+^ chondrocytes. We corrected the Results section and mentioned this fact. There were no other cell types found including traced perichondrial cells. Activation of caAlk2 somehow forces chondrogenesis and formation of spherical clusters instead of columns. This also shows that perichondrial cell are potent to produce chondrocytes when BMP pathway is activated. Therefore, there is no other neighboring tissue that bears activated ACVR1 and GFP. This is a pretty clear autonomous effect in the cartilage. Formation of spherical genetically traced bump and clumps is undeniable. Non-genetically traced cells (without caALK2 produced) behave in a normal way.

We expanded the description of results in the revised manuscript to discuss this particular detail.

*Similarly, all the PCP mutant mice were full knock-out, which makes it difficult to interpret the phenotype in cartilage.*

We state that the cartilage is perfectly fine at the level of clonal arrangements and we show that mesenchymal condensations are rather affected. This is all in the manuscript. We show that it is not a PCP pathway that directs the columnar oriented clonal behavior. Thus, the analysis of PCP mutants shows very important negative results: these full mutations do not influence clonal orientation and cartilage microgeometry. Only if the result would bring a change in the clonal structure – then we would need to go for conditional KOs.

*Gsa mutant mice are driven by Col2-Cre that is more chondrocyte specific, but the adjacent perichondrium were still labeled (Figure 5). In this case, the possible non-autonomous function of perichondrium could not be ruled out.*

We observe no significant phenotype at the level of general geometry. Only the cartilage thickness is affected, while lateral expansion is unperturbed. Since GSa is instrumental to achieve less cells comprising the column, it does not matter how exactly this effect is achieved (directly or indirectly). This experiment serves only the argument about uncoupling thickness control and the lateral expansion.

*5) Some other questions about Gsa mutant mice: The authors found that these more matured cells divided more slowly. But the clonal size appears similar (compare 5B and C, compare 5D and E). Were the mutant proliferative cells differentiated into hypertrophic cells? In the growth plate of long bones, these cells are larger than proliferative cells. Could these matured cells affect cartilage morphology in Gsa mutant? Since the lateral expansion of the tissue was not measured, how could the authors know cell division only affects tissue thickness?*

We introduced new analysis to answer this question. The shapes of the control and the GSa mutant are pretty identical despite differences in cartilage thickness (please see new in silico digital shape comparisons with GOM Inspect, Figure 8, new panel J). This strongly suggests the existence of uncoupled control of lateral expansion and transversal proliferation. The blue dots in the shape comparison image occur because of the specifically strong drops of thickness in these locations.

In locations that are proximal to synchondroses and inside of the forming inner ear we were observing more hypertrophic cells as compared to control. These hypertrophic cells appeared rather concentrated in few locations in the chondrocranium at the analysis stage. Thus, cell hypertrophy is increased in Gsa mutant, which is consistent with the previous findings. However, at the analysis stage, less cells comprising the column or cluster provide stronger effect of thickness than slight hypertrophy (hypertrophy is getting compensated). Importantly, multiple analyzed locations did not show the signs of hypertrophy at the time of our analysis and were much thinner as compared to the control because of the clonal size differences (see the thickness heatmaps and sections through the cartilage in Figure 8 for the proof of this reasoning).

*6) The progenitor role of perichondrium is uncertain. Based on the spatial distribution of clonally related cells, an alternative hypothesis is that the perichondrial cells are derived from chondrocytes.*

In the revised version of the manuscript we report the results of the live imaging of facial cartilage and regenerating salamander limb (new Figure 9 and Figure 10), where perichondrial cells generate chondrocytes incorporated into the main structure of the cartilage. Also, in our data one can see that in the regenerating limb very often chondrogenic mesenchymal progenitors generate both chondrocyte clone and spatially associated perichondrial cell. This is consistent with our EdU results and clonal genetic tracing associations throughout the manuscript. Still, the perichondrial layer is complex, specific markers are unknown, and often there is no clear border between the perichondrium and the cartilage. Thus, we cannot claim the exact nature of progenitor cells, and we do not do this claim in the revised version despite we obtained much better support for this hypothesis. In the new version of our manuscript we toned down this part and we only say that our data suggest that cartilaginous columns may originate from perichondrial layer during the cartilage growth.

*Because this lineage relationship is an important part of the paper, it is desirable to perform lineage-analysis using perichondrium specific Cre or time-lapse imaging to follow cell behaviors in the short-term. Time-lapse imaging on mouse craniofacial skeletons is technically possible (Romerim et al., 2014, Development).*

We addressed this comment experimentally. The imaging of the facial cartilage is extraordinarily hard. We performed it with only certain degree of success. We followed the advice of the reviewer and involved into collaboration the laboratory of Prof. Andrew Dudley (suggested by reviewer 2 by citing Romerim et al., 2014, Development). This collaboration resulted in better and comprehensive results (new Figure 9). Still, the total amount of cell divisions drops down tremendously in explant culture conditions as compared to the real embryo (three independent laboratories tried numerous conditions and cultivating approaches during this big collaboration). As a result, we consistently observe perichondrial (or flattened peripheral) cells producing chondrocytes in the cartilage and performing transversal cell divisions (despite rare events because of in vitro situation). It appeared that salamander regenerating limb represents a much better model system being naturally intact (after initial healing), transparent, integrated and with opportunity to perform lineage tracing with live imaging. We took advantage of this approach through collaboration with Elly Tanaka laboratory. The results showed that perichondrial cells indeed can give rise to the chondrocyte clones and that oriented clonal columns form during cartilage development similar to the situation in the facial cartilage (new Figure 9 and Figure 10, Figure 9—figure supplement 1, video). As we mentioned above, despite this experimental support, we decided to be cautious and rather suggest than claim lineage relationships between perichondrial and cartilage cells since we cannot experimentally address the amount of contribution.

*Reviewer #3:*

*[…] It is not clear why the authors have used PLP, Sox10 and Col2 cre mice. The Col2 appears to be enough to show the results. Are their different advantages of the different lines, or is there a benefit in targeting slightly different populations?*

There are different benefits of these different Cre lines since we wanted to explore clonal cell dynamics during formation of cartilage from early mesenchymal condensations and also we wanted to address how cells divide in the very early condensations. For this reason we needed to target neural crest stem cells, and therefore we recombined the confetti colors in the facial mesenchyme as early as E8.5 with the help of *PLP-CreERT2* and *Sox10-CreERT2*. However, later we wanted to explore cell dynamics in more mature cartilage, and *Col2-CreERT2* line appeared to be classical and working well. Still, at early stages of cartilage development and maturation it was not as efficient as *Sox10-CreERT2*. Thus, the reasons for using different Cre lines are purely technical. Importantly, all Cre lines show the same pattern and visualize the same process – this adds to the reliability of our study.

*The authors call their cartilages flat. I am not sure this is a good description of cartilages that are in complex 3D sheets. Perhaps their definition can be explained further. Do they mean a smooth surface?*

Yes, we mean the smooth surface and that these cartilage form sheets or curved planes. Typically, at the microscale these cartilages appear quite flat even if they eventually do bend on the macroscale. We agree with the reviewer that it is better to clarify the term, and following the reviewer’s hint we started to call these cartilages 3D sheets/planes or sheet-shaped cartilages in many places in our text. We hope it provides a more clear picture now.

*I cannot comment on the model presented from a mathematical point of view but the graphics are generally very good.*

We thank our reviewer for this comment. We did our best to perform good modelling!

*In the ACVR1 experiments clumps are shown at the periphery of the cartilages. This is a very nice clear result. In these mice are the cartilages shorter, as would be predicted if the cells were failing to intercalate successfully? In Image N there are quite a few GFP cells that might be expected to have altered the intercalation in this part of cartilage.*

ACVR1 experiment is indeed very clear and interesting. Still, the recombination efficiency (less than 5%) is not sufficient to ruin the geometry at really large scale. GFP+ clusters are quite few within the cartilages. This is the nature of this mouse model. We can see only very local disturbances in some locations where by chance we succeeded in higher recombination rate. We show the geometrical problems in 3D in one of those locations in Figure 6.

*The PCP mutants are interesting to illustrate the difference between initial patterning defects and problems with later scaling. Likewise the Gs-α experiments separate the role of integration from proliferation for controlling cartilage thickness.*

*In the early section of the text the authors suggest that the cartilage cells move in from the perichondrial layer, seeding the vertical columns. The origin of the new cartilage cells is an important point and is tested using dilution of H2B-GFP. I find this experiment the weakest in the paper as it does not clearly prove a perichondrium origin, but simply reiterates that the perichondrium and cartilage cells are clearly clonal.*

We completely agree with the reviewer and we removed the H2B-GFP part. It does not add anything new.

*The graph shows that the perichondrium appears to proliferate more than the cartilage cells but this is not showing point of origin. Time lapse would be a clear way of proving the origin of the cells.*

We completely agree with the reviewer and attempted to perform time lapse studies on thick slices of embryonic cartilages and regenerating salamander limb. These new data can be found in new Figure 9 and 10. Imaging of facial cartilage and regenerating limb shows that perichondrial cells can produce clones of chondrocytes. However, despite this is more direct evidence as compared to what we showed before, we decided to tone down the perichondrial origin of clonal columns since markers for this layer and its heterogeneity are unknown. We followed advices from the reviewers and decided to stay rather cautious. The readers can see the data and judge for themselves. In the revised version of the manuscript we do not claim, but only suggest that perichondrial cells provide for growth and regeneration by producing clones of chondrocytes.

*The last part of the paper deals with shape changes during later development. A key part is an analysis of bending but do the cartilages actually bend? From the Introduction I thought the nasal cartilage was simply scaling from an original pre-cartilage template with minor adjustments. The cartilage is not a sheet but already curved. If bending does occur this should be shown very clearly using real images, rather than the schematic shown. The sections of the nasal region in the supplementary data do not give much impression of bending, perhaps at the bottom of the capsule?*

*Confirmation of bending is necessary to confirm that modelling on flat plastic is relevant. The use of actual materials, however, is a very nice way to visualise morphogenetic movements, following from the work of Enrico Coen on plant morphogenesis.*

We do see how the new bends occur in the nasal capsule in transition from E13.5 to E14.5. The central groove becomes apparent and the lateral and anterior bends appear for the very first time. This allows the posterior bulge to form. All these bending elements are highlighted now in the real nasal capsule stage-dependent transformation outline in the new Figure 2. We made this point to be better explained in the new Figure 2 and figure legends thanks to the reviewer comments.

[Editors' note: the author responses to the re-review follow.]

*The manuscript is much improved but there are some remaining issues that need to be addressed by alterations in the text prior to acceptance, as outlined below:*

*1) With respect to whether chondrocytes function autonomously to control cartilage morphology, how could the authors exclude the possible roles of this layer? They argue that for an autonomous effect based on observing that cells with recombined GPF express Sox9 (Figure 6). However, there are two layers of perichondrium: and inner layer that expresses Sox9 and an out layer that expresses collagen type I. Could this outer layer cause a non-autonomous effect in cartilage shaping? The outer layer perichondrium also contains recombined cells in both confetti mouse (Figure 6) and caALK2 mouse (Figure 6), both of which are driven by Sox10 promoter. Therefore, the possibility of non-autonomous effects should be discussed.*

The possible non-autonomous effects are now discussed: “In addition to this, the perichondrium might mediate non-autonomous effects in the cartilage in case of cAlk2 and GSα experiments. Genetic tracing shows that some perichondrial cells always recombine with *Sox10- PLP-* and *Col2-CreERT2* lines, and, in case of functional experiments, may indirectly control some evens in more mature layers. Also, it is not clear how the fine border of the cartilage is set, and whether the perichondrial layer may play a key border-setting role during development and regeneration. This should be investigated further.”

*2) Kudos to the authors for trying very hard to do live imaging to test the lineage relationship between chondrocytes and perichondrium. However, the "n" (Figure 9) is very low. Given that the total number of cell division drops in organ culture, such that they cannot perform statistical analysis, they should leave out this data.*

We agree and we removed the data.

*3) In Figure 9, the authors use salamander regenerating limb as a tissue model for live imaging. However, cell behaviors could be very different during normal development versus regeneration. In addition, it previously has been shown that during cartilage regeneration, perichondrium cells can regenerate chondrocytes (Srour MK et al., Natural large-scale regeneration of rib cartilage in a mouse model, 2015). Therefore, these data may not be relevant to the story and the authors may want to leave it for a different paper.*

We agree and we reorganized the figure and the text so it better reflects the discussion of clonal patterns during growth of rod-shaped cartilages rather than perichondrial cells as a source of growth.

*4) The paper would profit from some rewriting. Parts of the paper are not well organized; the Summary and Introduction are too long and some of the contents seem to be misplaced; there are many grammatical mistakes.*

We shortened the Summary and Introduction, reorganized the text to make it better ordered and we also fixed all grammar mistakes. Finally, we performed the large-scale language correction being assisted by native speakers.